# Response Mechanism of Plants to Drought Stress

**Xinyi Yang, Meiqi Lu, Yufei Wang, Yiran Wang, Zhijie Liu and Su Chen ***

State Key Laboratory of Tree Genetics and Breeding, Northeast Forestry University, Harbin 150040, China; yangxinyi@nefu.edu.cn (X.Y.); lumeiqi0408@163.com (M.L.); wangyufei@nefu.edu.cn (Y.W.); wangyiran@nefu.edu.cn (Y.W.); hjdyyy69@163.com (Z.L.)
* Correspondence: chensu@nefu.edu.cn

**Abstract:** With the global climate anomalies and the destruction of ecological balance, the water shortage has become a serious ecological problem facing all mankind, and drought has become a key factor restricting the development of agricultural production. Therefore, it is essential to study the drought tolerance of crops. Based on previous studies, we reviewed the effects of drought stress on plant morphology and physiology, including the changes of external morphology and internal structure of root, stem, and leaf, the effects of drought stress on osmotic regulation substances, drought-induced proteins, and active oxygen metabolism of plants. In this paper, the main drought stress signals and signal transduction pathways in plants are described, and the functional genes and regulatory genes related to drought stress are listed, respectively. We summarize the above aspects to provide valuable background knowledge and theoretical basis for future agriculture, forestry breeding, and cultivation.

**Keywords:** drought stress; osmotic regulation; LEA protein; ROS; signaling; drought-responsive gene

## 1. Introduction

Drought is one of the most important factors restricting agricultural production, which seriously affects crop yield [1,2]. Moreover, as one of the main restraining factors in the process of plant growth, drought can hinder plant respiration, photosynthesis, and stomatal movement; thus, affecting plant growth and physiological metabolism. In response to drought stress, plants activate their drought response mechanisms, such as morphological and structural changes, expression of drought-resistant genes, synthesis of hormones, and osmotic regulatory substances to alleviate drought stress. To better reveal the mechanism of drought resistance of plants, based on a lot of previous work, we summarized the status quo and progress of studies on the morphological structure, physiological and biochemical mechanism changes, internal signal transduction system, and molecular regulation mechanism of plants under drought stress in recent years. Under drought conditions, plants sense water stress signals and produce signal molecules, such as abscisic acid (ABA), $Ca^{2+}$, inositol-1, 4, 5-triphosphate (IP3), cyclic adenosine 5′-diphosphate ribose (cADPR), NO, etc., and directly or indirectly lead to the morphological and physiological changes of plants through signal transduction. Indirectly, drought stress signals induce the expression of downstream genes. Functional gene products, such as proline (pro), glycine betaine (GB), soluble sugar (SS), late embryogenesis abundant (LEA) proteins, and aquaporin (AQP) can be involved in plant metabolism and, thus, affect plant state. Regulatory gene products, such as calcium-dependent protein kinases (CDPKs), mitogen-activated protein kinases (MAPKs), HD-zip/bZIP, AP2/ERF, NAC, MYB, and WRKY can cause changes in plant morphology or physiology by regulating signal trans-

duction pathways or acting as transcription factors to regulate the expression of downstream genes, and further enable plants to successfully survive in the arid environment (Figure 1).

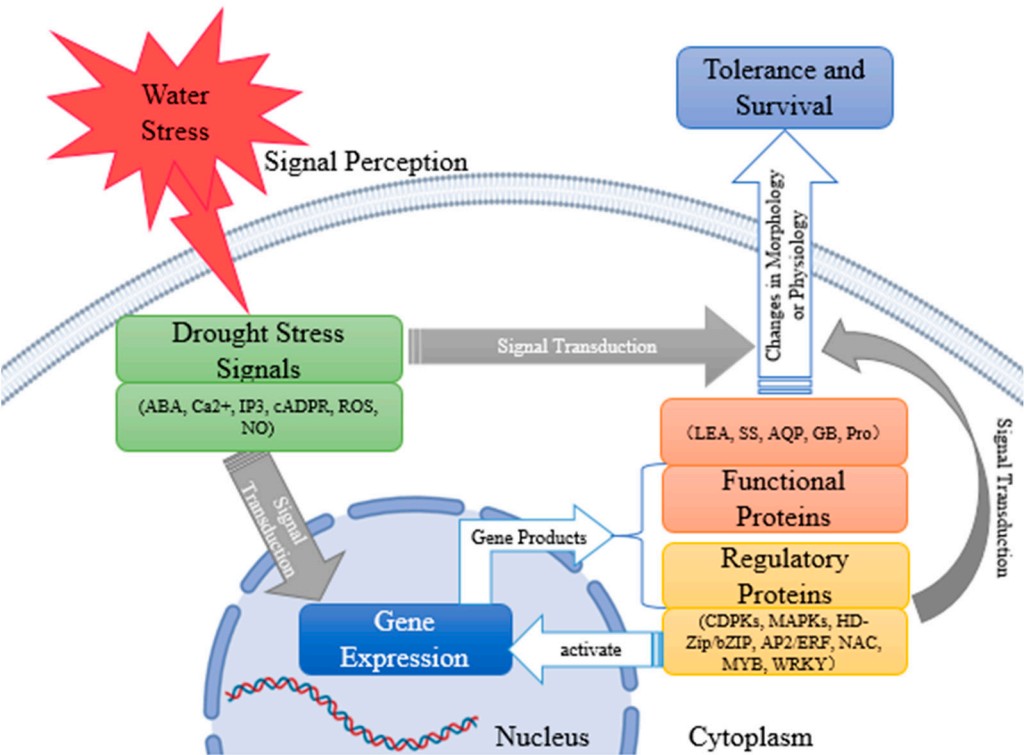

**Figure 1.** The process of plant drought-tolerance development.

We will elaborate from the following four parts. The first is the effect of drought stress on the external morphology and internal structure of plants. The second part elaborates the physiological and biochemical responses from the perspectives of osmotic regulation metabolism, drought-induced protein metabolism, and reactive oxygen metabolism. Here we summarize some important drought-regulating substances and we also briefly summarize the generation and scavenging process of reactive oxygen species (ROS). The third part is the signal transduction pathway in plants. We describe common signals in detail and elucidation of intracellular signal transduction pathways. The fourth part is about the molecular regulation mechanism of plants. From the perspective of genes, the anabolism and regulation mechanisms of osmotic regulation-related substances, drought-induced proteins, signaling path-related substances, and transcription factors are summarized respectively. All of the advances indicate that it is of great significance to study the effects of drought stress on plants and explore the mechanism of drought tolerance.

## 2. Effects of Drought Stress on Plant Morphological Characteristics

When plants are subjected to drought stress, they will first respond to changes in external form and internal structure. The most significant effect of water loss is that the plant grows slowly and even dies. Studies have shown that plants under abiotic stress can adapt to changing environmental factors through phenotypic plasticity. Therefore, under the influence of the environment, xerophytes have formed certain morphological characteristics in the process of evolution, and adapted themselves to drought in their ontogenetic development under these characteristics. The drought-resistant plants have morphological and structural characteristics that were adapted to the arid environment in terms of leaves, stems, roots, and so on.

### 2.1. Drought Stress and the External form of Plants

The obvious symptoms of water deficit during the vegetative period are plant height decreased, leaf wilting, number and area of leaves changed. Plant height, severely affected by drought, is closely related to cell enlargement and leaf senescence. The decrease in plant height is mainly due to decreased cell expansion, increased leaf shedding, and impaired mitosis under drought conditions. Some studies have reported that plant height of lily [3], maize [4], cane [5], and rice [6] decreased significantly under drought stress. In addition to the changes in plant height, different organs of plants also differ significantly in morphology. As an indicator of the degree of water shortage in direct response, leaves are the main organs for plant assimilation and transpiration. Plant leaves generally adopt smaller leaf areas, larger leaf thickness, and higher leaf tissue density to adapt to drought [7]. The change of leaf area, which directly affects plant photosynthesis and yield, is one of the most easily observed features of plant leaves under drought stress. Previous studies have shown that the main reasons for the change of plant leaf area are the leaf turgor pressure, canopy temperature, and availability of photoassimilates [8]. Under the condition of drought, the leaf turgor pressure and the rate of photosynthesis of plant leaves decrease, which leads to the decrease of leaf area [9]. For morphological responses, *Prunus sargentii* and *Larix kaempferi* experienced a significant decrease in leaf size, respectively in leaf width and length under drought conditions [10]. Furthermore, *Maclura pomifera* [11], *Oryza sativa* [6], *Triticum aestivum* [12], *Lens culinaris* [13], *Dracocephalum moldavica* [14] all showed an obvious decrease in leaf area under drought stress. However, different plants have different responses to drought stress, such as sugarcane leaves showed marginal elongation under drought stress [5]. Another easily observed leaf morphology phenomenon is leaf rolling, for the loss of the potential pressure due to water loss from the upper epidermis of the leaf when plants are short of water. Under drought stress, the flag leaf of wheat would be severely rolling [15]. In a xerophytic environment, conifers have thick horny film, and their wilting and rolling motion can resist direct sunlight to improve their water retention [16,17].

Apart from leaves, plant roots, as organs that directly absorb water, also play a significant role in drought stress [18]. Developed roots can help plants to fully absorb and utilize the water stored in the soil so that plants can survive the drought period [19]. What is more, researches have shown that water is the main environmental factor affecting the development of plant roots [20]. Therefore, the morphological changes of plant roots in arid areas are particularly important. Root system configurations such as root hair, root branches, and root density can significantly affect the water deficiency of plants. Drought stress can inhibit the development of cotton seedlings, promote the elongation and thinning of fine roots, shorten the life of fine roots with different diameters, promote the elongation of root hairs, and accelerate their death [21]. *Cunninghamia lanceolate* can increase root complexity and elongation, reduce root branching angles, leading to steeper and deeper roots system to adapt to drought stress [22]. Maize treated by drought stress obtains more water from dry soil by reducing lateral root branch density, making axial root elongation and rooting depth larger [23]. Water also had a certain effect on the distribution of plant roots. Soybean, field pea, and chickpea were sensitive to the soil moisture content of biomass decreased more than the root, leading to a higher root/shoot ratio of soybean [24]. Drought can also affect the external morphology of plants in other ways, such as the average internode length of sugarcane increased by 39.02% after drought treatment in an early vegetative stage, and drought stress would destroy the full root structure [25]. In addition to the above characteristics, the root to stem ratio of plants also changes. The shoot and root biomass of soybean decreased significantly under drought stress. Under the condition of water restriction, the height, leaf size, and stem girth of maize plants decreased significantly [26].

## 2.2. Drought Stress, the Internal Structure, and Physical Property of Plants

In addition to the external form, the internal structure of plants also changed. There is a developed cuticle in the outer wall of the leaf epidermis. The cuticle is a kind of lipid membrane, which can reduce the loss of water to the atmosphere and acts as a barrier for plant water evaporation. The thick cuticle can improve the plant's energy reflection and reduce transpiration, thus enhancing the plant's drought resistance. The cuticular lipids content of *Arabidopsis thaliana* leaves increased significantly under water shortage treatment. The increase of epidermal wax per unit area under drought stress was mainly due to the increase of wax alkanes. Moreover, the water deficit increased the total amount of cutin monomers, changed the proportion of the cutin monomers amount, increased the thickness of the leaf cuticle, and the accumulation of osmium in the plant cuticle [27]. Tea leaves improve drought resistance through increasing wax coverage, cuticle thickness, and osmiophilicity [28]. In addition, plant leaves tend to increase mesophyll palisade tissue, decrease spongy tissue, increase the number of cell layers, but decrease the volume and shorten the intercellular space to adopt drought [29]. Stomatal development is another important index related to water stress. The drought process appeared to increase stomatal length, stomatal width, stomatal density, and stomatal opening. The reduced stomatal density of *Hordeum vulgare* leaves could increase its tolerance to water stress [30]. Apple cultivars, which exhibited significantly thicker cuticle, longer palisade cells, and thicker spongy parenchyma had superior drought tolerance [31]. In the observation of micromorphology of blackberry after drought treatment, it was found that with the extension of stress time, the morphology of leaf epidermis cells underwent a series of expansion changes. Moreover, the walls of the epidermal cells and spongy tissue cells of the leaves thickened with the duration of drought. Especially after treatment for a period of time, the spongy tissue cells were obviously compressed and filled with sclerenchyma [32]. What is more, the degree of lignification and channeling tissue on the epidermis had a great influence on the drought resistance of the plant. The study found that plants with water deficits had lower levels of lignin in their leaves than those with adequate water [33]. The xylem of the stems and roots of the stress-treated plants was thicker than that of the normal rapeseed plants. In addition, drought stress reduced the vessel's inner diameter and increased the number and inner diameter of root vessels [34]. Fresh and dry weights are also significantly reduced under water deficit conditions [35]. In the study of *Matthiola incana*, the relative water content did not change significantly with the increase of drought stress, but plant height, stem fresh weight, stem dry weight, root fresh weight, and root dry weight all decreased significantly [36]. Besides, water stress had significant effects on the essential oil content and essential oil composition of Rosemary. With the decrease of soil water content, stalk length, fresh weight, and fresh and dry weight of root decreased. At the same time, the content of essential oil also presents the trend of first rising and then falling [37].

## 3. Effects of Drought Stress on Plant Physiological and Biochemical Characteristics

When plants are subjected to drought stress, a series of changes will occur in their appearance, leading to a series of physiological and biochemical changes in plants. For example, the changes in photosynthesis, osmotic regulatory substances, drought-induced proteins, and antioxidant enzymes all reflect the different degrees of influence of plants under drought stress.

### 3.1. Photosynthetic Capacity

Photosynthesis is one of the main processes affected by water stress. Leaf photosynthetic products are the material basis of plant growth. The net photosynthetic rate directly reflects the material productivity per leaf area. Therefore, theoretically speaking, it is a reliable index to measure the biological production level of plants. The photosynthetic rate and transpiration rate decrease with the decrease of soil relative water content. Previous

studies have shown that the decrease of photosynthetic rate under drought stress is the result of stomatal limitation and non-stomatal limitation. The stomatal limitation was the main factor of photosynthetic rate decrease under mild drought. However, under severe drought conditions, non-stomatal factors were the main reason for the decline of the photosynthetic rate. When water is deficient, it will lead to the decrease of photosynthesis directly through decreasing $CO_2$ availability resulted in diffusion limitations of the stomata and the mesophyll [38]. Stomatal closure limits leaf absorption of $CO_2$ and prevents transpiration water loss due to turgor pressure and/or reduced water potential. In a study by Victor Santos et al., they pointed out that photosynthesis in the canopy of the central Amazon forest decreased by 28% in the dry season and by 17% in the undergrowth, compared with that in other seasons in 2015. They further suggested that the reduction in photosynthesis was only related to the closure of stomata in trees in the canopy and undergrowth [39]. It was also found in wheat that drought decreased stomatal conductance, increased stomatal resistance, and decreased photosynthetic rate and transpiration rate [40]. However, with the increase of water deficit, non-stomatal factors began to play an important role. At this time, the potential photosynthetic $CO_2$ assimilation rate decreases, which cannot be eliminated by increasing the external $CO_2$ concentration. The decrease of photosynthesis, which is dominated by non-stomatal factors, is related to the decrease of activity or component content of many important processes related to photosynthesis. For example, Ribulose-1,5-bisphosphate (RuBP) content and activity, as well as apparent quantum yield, play a very important role in the photosynthetic assimilation process. In the study by Carmen Gimenez, it was found that there was an obvious S-shaped curve relationship between the photosynthetic rate and RuBP in sunflower leaves, suggested that the reduction of photosynthetic rate was to some extent restricted by RuBP content [41]. Dhammika's study on tobacco also confirmed that RuBP synthesis is limited under water stress due to inhibition of the activity of synthetic enzymes [42]. Another substance that is important for plant photosynthesis under water stress is the enzyme Ribulose-1,5-bisphosphate carboxylase/oxygenase (RuBisCo). The activity of the RuBisCo enzyme had no significant change or was less affected under mild water shortage, but decreased under severe drought. In addition, changes in photochemical and biochemical processes such as electron transfer rate decrease and photophosphorylation are also observed. The direct manifestation of these changes is the occurrence of "photoinhibition". With the increase of drought intensity, the net photosynthetic rate, transpiration rate, and stomatal conductance of cotton decreased [43]. Ma Ping et al. studied the effects of drought on photosynthesis in apples. Soil relative water content (SRWC) decreased from 87% to 24% within 15 days after the irrigation treatment was stopped, while leaf relative water content (LRWC), net photosynthetic rate (Pn), and stomatal conductance (GS) all showed a decreasing trend. Moreover, they noted that the photochemical reaction was only slightly downregulated under severe drought conditions. With the intensification of drought conditions, the activity of RuBisCo decreased significantly, and the actual efficiency of photosystem II ($\Phi$PSII) decreased [44].

The chloroplast is the site of photosynthesis in green plant leaves, which mainly uses chlorophyll to absorb, transfer and transform light energy. Chlorophyll is continuously metabolized in plants, closely related to photosynthesis and yield formation of plants. As the most important and effective pigment in photosynthesis, chlorophyll can reflect the growth status of plants and the degree of stress. Chlorophyll content tends to decrease under drought stress and the ratio of chlorophyll "a", "b", and carotenoid was changed, thus, in turn, causes changes in photosynthetic function [45]. The reason for the decrease of chlorophyll content in leaves may be the degradation of chlorophyll directly caused by drought. Drought stress could significantly reduce the contents of chlorophyll a, chlorophyll b, and total chlorophyll in chickpea during vegetative growth and anthesis [46]. In the study of 13 durum wheat native varieties from Iran and Azerbaijan, it was found that different wheat varieties had different responses to drought stress. The chlorophyll level of susceptible wheat cultivars decreased significantly under drought stress, while the

chlorophyll content of resistant wheat cultivars was still maintained [47]. In Chinese cork oak (*Quercus variabilis*) seedlings, chlorophyll a, chlorophyll b, carotenoids (Car), and total chlorophyll contents were significantly decreased at 40% and 20% field capacity, despite there was no significant change in Chl a/Chl b and Car/Chl ratios [48]. The total chlorophyll content and the ratio of Chl a/Chl b in oil palm were significantly decreased under water stress [49]. However, not all plants show reduced chlorophyll content under drought stress. Soheila pointed out that chlorophyll content in borage increased at lower irrigation levels, mainly due to lower leaf area index and more radiation interception [50]. Besides, drought makes it difficult for plants to absorb nutrient elements and causes symptoms of deficiency of elements, which is also manifested as decreased chlorophyll content. Changes in plant pigments lead to the color of the plant changed into yellowish-brown when they suffer from drought. From the point of view of drought resistance, plants with high chlorophyll content generally have stronger drought resistance.

The pathways of $CO_2$ assimilation in photosynthesis can be divided into the $C_4$ pathway, $C_3$ pathway, and Crassulacean acid metabolism (CAM) pathway. Under drought stress, the $C_4$ pathway was significantly superior to the $C_3$ pathway. The leaves of $C_4$ plants have a typical Kranz wreath structure and water use efficiency (WUE) is significantly higher than that of the $C_3$ pathway. Under the condition of water shortage, the $C_4$ pathway can assimilate $CO_2$ to produce more organic matter, which is conducive to the plant to resist early drying. The stomata of CAM plants open at night, absorb $CO_2$, and form malic acid catalyzed by phosphoenolpyruvate carboxylase (PEPC), which is stored in vacuoles. Stomatal closure during the day, MAL decarboxylation gives off $CO_2$. Because CAM fixes $CO_2$ by stomatal opening at night, transpiration loss during the stomatal opening in the day is avoided, and the contradiction between stomatal transpiration and $CO_2$ absorption under drought stress is solved. Different species have different assimilation pathways and different environmental conditions can significantly change the carbon metabolism pathways of plants. That is to say, changes in growth and development level, growth conditions, nutritional status, and biological regulators can lead to a mutual transformation of $CO_2$ fixation pathways in plants. Increased ABA content in plants under drought conditions promotes the operation of the $C_4$ pathway. Similarly, the $C_3$ pathway can also be transformed into the CAM pathway [51]. Winter's studies on different varieties of orchids and *Mesembryanthemum crystallinum* L. (Aizoaceae) have shown that some species with highly flexible photosynthetic phenotypes have changes in assimilation pathways when the external environment changes. They operate in $C_3$ mode when not stressed, or in CAM mode when drought or salinity stressed [52–54]. Milton Garcia support that the $C_3$-CAM shift is present in the cactus seeding process. Ideas are put forward that there is a facultative component of CAM expression in the cactus. Shortly after germination, the expression of $C_3$ photosynthesis can promote plant growth when there is sufficient water. Facultative CAM components can accelerate the development of constitutive CAM and contribute to plant survival in water-deficient environments [55].

### 3.2. Osmotic Regulation Metabolism

Osmotic regulation is an important way for plants to reduce osmotic potential and resist adversity stress under water stress. When plants are subjected to drought stress, osmotic regulation can be realized in three ways, namely, the decrease of intracellular water, the decrease of cell volume, and the increase of cell contents. These three pathways coexist in plants, but not all plants have osmotic regulation. Osmotic regulation is generally considered to be the active regulation of cells to reduce osmotic potential by increasing solute. Its initial effect is to reduce the free energy of water bound inside the cell, maintain the difference of water potential inside and outside the cell, and enable the cell to absorb water under the condition of lower external water potential. Thus maintaining the turgor pressure required for cell growth [56]. Osmotic regulation can maintain stomatal conductance to moderate water deficit by maintaining turgor pressure. It helps to keep the content of $CO_2$ in mesophyll intercellular space at a high level so as to avoid or reduce

the photosynthetic inhibition on photosynthetic organs. Osmotic regulation can maintain normal or minimize damage to biochemical, physiological, and morphological processes related to cell growth, stomatal opening, and photosynthesis during environmental stress. The osmotic regulating substances in plants mainly include organic osmotic regulating substances and inorganic ions entering from the external environment. Organic osmotic regulating substances, such as amine compounds (glycine betaine and polyamines), amino acid compounds (proline), and trehalose, fructan, mannitol, and other compounds, play a major role in regulating the osmotic type of cytoplasm. These substances are usually of small molecular weight, highly soluble, and have little toxicity to cells. They can maintain the normal osmotic pressure level, protect the protein activity and cell membrane structure, etc. The osmotic regulation of inorganic ions is closely related to the ion pump. For example, the $Na^+$, $K^+$, $H^+$ pump can regulate the concentration of inorganics inside and outside the cell, thus changing the osmotic potential of the cell. At the same time, the change of inorganic ion concentration will cause a change in cell morphology and function. Suomin Wang et al. proposed that $K^+$ and free proline accumulation played an important role in drought adaptation of xerophytic plants and $Na^+$ accumulation is one of the most effective strategies for succulent xerophyte to adapt to drought [57]. At present, there are more studies on osmotic regulation substances such as proline (Pro), soluble sugar (SS), glycine betaine (GB), etc. Some studies have found that Pro accumulation is a protective measure taken by plants to resist drought stress [58]. When PEG concentration was 30%, Pro content in rice increased significantly [6]. Under drought stress conditions, osmotic regulation substance content increased, which was positively correlated with plant stress resistance. However, the variation range of osmotic regulation substance was different among different species. By decreasing soluble sugar, polysaccharide, and fructose contents and increasing proline, glucose, and trehalose contents, Lanzhou lily can improve its resistance to drought stress by changing the contents of osmotic regulation, and secondary metabolites [3]. The contents of soluble carbohydrates sucrose, glucose, and fructose in *Maclura pomifera* increased at the initial stage of drought stress but decreased after 22 days of severe drought stress. In addition, the affinity of osmotic substances proline and mannitol increased significantly under drought stress [11]. In the study of Farooq et al., the contents of proline, glycine betaine, total soluble carbohydrate, and sucrose were significantly increased due to drought stress in several pistachio genotypes [59].

As an osmotic regulating substance, proline (Pro) is preferentially stored in plant vacuoles. When the cell is subjected to osmotic stress, Pro is transported to the cytoplasm, and the osmotic potential is reduced by increasing the concentration of the cytoplasm so that the cell can still absorb extracellular water under the condition of low osmotic potential; thus, maintain the cell protoplasm and the external environment of osmotic balance [60]. Pro has a strong ability to hydrate, so it can also play a protective role in cell structure. In the event of plant injury, Pro interacts with proteins to form a hydrophobic skeleton to stabilize and protect biological macromolecules and cell membrane structures. Pro is also a variety of free radical scavengers [58]. Pro can reduce the oxygen damage caused by stress through chelating singlet oxygen and hydroxyl radical. Another way of Pro to remove ROS is to stimulate the activity of POD, catalase (CAT), superoxide dismutase (SOD), polyphenol oxidase (PPO), and other enzymes in plants. Under the stress of adversity, Pro can bind to proteins to form a protective film with water molecules on the surface of proteins. The formation of a protective membrane restrains the flow of water to the outside of the cell and reduces the loss of water. Moreover, the protective membrane has a good protective effect on proteins and other biological macromolecules, maintaining the high structure and activity of biological macromolecules. For denatured proteins under the stress of adversity, Pro can improve the hydrophilicity of denatured proteins after combining with it. It keeps the dissolved state of the denatured proteins so as to avoid the agglutination of the denatured proteins interfering with the metabolic activities of the cells. Therefore, Pro is an important osmotic regulating substance.

Glycine betaine is a water-soluble substance with amphoteric characteristics. As an effective non-toxic osmotic regulator, it can bind to both hydrophilic and hydrophobic regions of biological macromolecules such as enzymes. Drought stress can cause the accumulation of glycine betaine and it can improve the drought-resistant ability of plants [61], which have been proven in sunflower [62], wheat [63], barley [64], pepper [65], *Axonopus compressus* [66], etc. The application of glycine betaine can effectively improve the osmotic regulation ability, stomatal conductance, and carboxylation efficiency of $CO_2$ assimilation so as to promote photosynthesis [67]. In other words, under drought stress, glycine betaine can stabilize the structure and properties of biological macromolecules, such as the key enzymes of the dicarboxylic acid cycle, terminal oxidases, and the photosystem, etc., which have important physiological significance in maintaining normal respiration and photosynthesis of plants [58]. Soluble sugar (SS) is an important energy and carbon source in the organism and participates in many processes of plant life metabolism. The soluble sugar in general plants includes glucose, fructose, sucrose, and other carbohydrates. The accumulation of soluble sugar can reduce the water potential of cells and improve the ability of plants to absorb water and retain water. In addition, most osmotic regulators fail to protect proteins and biofilms with further water loss under severe drought stress. Only soluble sugars can take the place of water molecules and form hydrogen bonds with proteins to maintain the specific structure and function of proteins. Moreover, the increase of soluble carbohydrates between biofilms can avoid the direct collapse of the biofilm system.

However, osmotic regulation also has limitations. The improvement in drought resistance of plants is only temporary. Moreover, it has a very limited effect on plant drought tolerance. If drought stress is severe, the turgor pressure of plants cannot be maintained. The effects of drought are present even within the range of osmotic adjustment of water potential. Osmotic regulation can only alleviate drought damage of plants to a certain extent.

### *3.3. Drought-Induced Proteins*

Drought-induced proteins are newly synthesized proteins in plants under drought stress, which play a protective role in plant adaptation to stress and can improve plant drought tolerance. Drought-induced proteins can be divided into two categories according to their functions: (1) functional proteins, which play a direct protective role in cells, mainly include ion channel proteins, LEA proteins, OSM proteins, and metabolic enzymes, etc. (2) Regulatory proteins, including protein kinases, phospholipase C, phospholipase D, G protein, calmodulin, transcription factors, and some signaling factors, are involved in signal transduction or gene expression regulation in water stress and play indirect protective roles. Three important drought-inducible proteins, LEA, AQP, and dehydrin, are highlighted below.

### 3.3.1. Late Embryogenesis Abundant Protein

Late embryogenesis abundant (LEA) protein is a dehydrating protective protein enriched in the late stage of seed embryo development. LEA protein is rich in lysine and glycine, most of which are between 10 and 30 kD, and a few of which are above 30 kD. LEA is a large family of proteins, with more than 50 of them found in Arabidopsis alone [68]. It is regulated by plant development stage, ABA and dehydration signal, etc., and can be expressed in many tissues and organs of plants, with high hydrophilicity and thermal stability. The ability to capture enough water into cells is closely related to the dehydration tolerance of plants and the protection of tissues from water stress. Most LEA proteins do not have a stable secondary structure, but they may acquire an $\alpha$-helix structure after drying [69]. LEA protein can participate in the process of crop resistance to environmental stress and plays a key role in this process, which is closely related to its amino acid composition and structure. Most LEA proteins contain a high proportion of polar amino acids, which makes them highly hydrophilic. What is more, most LEA proteins contain

some conserved sequences. These sequences can form high helical folding under stress conditions, and such structure may have a hydrophobic effect with the membrane system of some denatured proteins. By stabilizing lipid membrane or functional proteins, a large amount of water loss can be prevented, thus reducing the influence of the external environment on intracellular metabolism [70]. In addition, there is a dynamic equilibrium between random conformation and $\alpha$-helix in the dissolved state of LEA protein, which is also one of the reasons that LEA protein can participate in the resistance of crops to environmental stress [71].

One of the important functions of LEA protein in response to stress such as drought is its ability to scavenge ROS. Plants will produce a large number of reactive oxygen free radicals under adverse conditions, which have strong oxidation properties and can damage cell membranes and proteins, etc. Therefore, the scavenging of reactive oxygen free radicals becomes an important protection mechanism of plants under adverse conditions. Hara et al. found that the dehydrin CuCOR19 in *Citrus reticulata* can scavenging hydroxyl radicals and hydrogen peroxide and reduce the damage of reactive oxygen free radicals to plants [72]. Dean et al. found that glycine, lysine, and histidine were vulnerable to free radical attacks, and the total contents of glycine, lysine, and histidine in citrus dehydrin CuCOR19 were up to over 40%. Therefore, LEA protein could consume part of free radicals through three amino acids and play a protective role on plants. Because the LEA protein has no obvious secondary structure, the oxidation of some amino acids is obviously not sufficient to completely destroy its function [73]. Lea protein can prevent the loss of water by binding to the lipid membrane through the $\alpha$ helix structure, so it has the function of binding to the membrane to stabilize the membrane. For example, in corn, dehydrin DHN1 can bind to vesicle membrane containing acidic phospholipids, and its helicity will be significantly enhanced, indicating that functional conformational changes of DHN1 have occurred at the membrane interface, which may be related to dehydrin maintaining the stability of vesicle membrane and other intimal structures under adversity [74]. Hara et al. also found that overexpression of *CuCOR19* in tobacco could inhibit lipid membrane peroxidation of tobacco [75]. Thalhammer et al. found that the Arabidopsis LEA proteins COR15A and COR15B could bind to lipid membranes under drought conditions and play a protective role in lipid membranes [76]. In addition to scavenging reactive oxygen free radicals and maintaining the stability of the intimal system, LEA protein can be used as a cryo-protectant and metal ion protectant to participate in a wide range of stress. It was found that most of the LEA proteins of the dehydrin family play an important role in the process of cold resistance in plants. The accumulation of dehydrin gene *WCS120* was significantly correlated with its survival rate during winter in wheat [77]. Overexpression of *DHNS* in *Arabidopsis thaliana* showed that the cold tolerance of *Arabidopsis thaliana* was improved [78]. There is no domain associated with metal ion binding in dehydrin, but the proportion of histidine in dehydrin is high. It is speculated that the high proportion of histidine in dehydrin makes dehydrin have the ability to bind metal ions. It was also found that His-X and His-X3-His (X is an arbitrary amino acid) structures existed in many dehydrators, and further study confirmed that the ability of these domains to bind metal ions was significantly higher than that of other amino acids [79]. However, in the study of *Ricinus communis*, it was found that there was no correlation between the ability of dehydrin to bind metal ions and histidine content, which indicated that dehydrin had a more complex mechanism of binding metal ions [80].

Similar to soluble sugar in cells, LEA proteins with high hydrophilicity bind a large number of water molecules, allowing plants to maintain normal metabolism without damaging cells even in the event of severe dehydration [81]. The facultative $\alpha$-helix structure formed by LEA protein interacts with the cell membrane under dehydration conditions, making the cell membrane maintain a relatively stable state even under dehydration conditions, thus preventing water loss [71]. In addition, most of the *LEA* genes have ABA response elements in their promoter regions, so the increase of endogenous ABA content in plants under drought conditions can also lead to the increase of *LEA* gene expression

[82]. Under drought stress, LEA protein is also a good enzyme protectant. For example, the *LEA* genes of *Boea hygrometrica*, *LEA1*, and *LEA2*, were transferred into tobacco (*Nicotiana tabacum* L.) to obtain transgenic tobacco. It has found that the transgenic tobacco superoxide dismutase (SOD), peroxidase (POD), and photosynthetic system II related enzyme activity was increased. Moreover, the water content of leaves also increased. Thus, the stability of its protein is enhanced [83]. Overexpression of some *LEA* genes can improve the drought resistance of plants. Studies have confirmed that the overexpression of wheat *LEA* gene *TaLEA3* into *Leymus chinensis* can improve drought resistance [84].

3.3.2. Dehydrin

Dehydrin, a member of the Lea-II family with a molecular weight of 9~200 kDa, is a drought-induced protein widely found in higher plants. It is produced during late embryogenesis and responds to low temperature and exogenous ABA, or typically accumulates in dehydration stressed plants under drought, salt, and extracellular freezing. Dehydrin is rich in glycine and lysine and lacks cysteine and tryptophan. It is highly hydrophilic. In addition, dehydrin is a heat-stable protein that remains stable in boiling water and is thought to play an important role in protecting cells from damage caused by cell dehydration. An important structural feature of dehydrin is that it has three conserved regions: K, S, and Y fragments. The K fragment consists of 15 amino acids (EKKGIMD-KIKEKLPG) and is rich in lysine. The K fragment is usually located at the C end of the protein sequence and can form the amphipathic $\alpha$-helix, which is the important structural basis of its hydrophilicity [70]. The S fragment is composed of a series of serine residues, and phosphorylation of the S fragment has been shown to enable dehydrin to enter the nucleus guided by signal peptides [85]. The conserved sequence of fragment Y is (T/V) D (E/Q) YGNP, located at the N-terminal of the dehydrated protein. The fragment Y is homologous to the nucleic acid binding sites of some bacterial and plant molecular chaperones. In addition, dehydrin has some conservative less rich in polar amino acid of $\Phi$ fragments and approved a similar nuclear localization signal (NLS) sequence [86]. According to the number of K, S, and Y fragments, the plant dehydrin gene family can be divided into five subfamilies: $K_n$, $SK_n$, $Y_nSK_n$, $Y_nK_n$, and $K_nS$.

In an aqueous solution, dehydrin forms the largest amount of hydrogen bonds with neighboring water molecules, while the proportion of external hydrogen bonds is very low, which does not form the hydrophobic core required for folding protein. Therefore, the dehydrin protein presents an unstructured and disordered protein form without a fixed three-dimensional structure. However, when the microenvironment around the dehydrin protein changes, the conformation of the dehydrin protein also changes. In the dehydrated state, the K fragment forms an $\alpha$-helix type conformation in which the negatively charged amino acids lie on one side of the helix, the hydrophobic amino acids on the other, and the positively charged amino acids lie on the polar nonpolar interface. The $\alpha$-helix with both hydrophilic and hydrophobic properties can interact with the dehydrated surfaces of other proteins or biofilm surfaces [87]. Therefore, dehydrin plays a stabilizing role in protecting the membrane system. Stress often dehydrates plant cells, destroys the hydration protection system on the surface of membrane lipid bilayers, reduces the space between membrane lipid bilayers, and causes membrane fusion and severe destruction of membrane structure. The amphiphilic $\alpha$-helix formed by the K fragment in the dehydration condition enables the dehydrin to participate in the hydrophilic and hydrophobic interactions. Due to its high hydration ability, dehydrin binds with membrane lipids to prevent excessive loss of water in cells, maintain the hydration protection system of membrane structure, prevent the decrease of membrane lipid bilayer spacing, and thus prevent membrane fusion and the destruction of biofilm structure [88]. Dehydrin also protects the protein. The amphipathic $\alpha$-helix formed by the K fragment can bind the dehydrin to the hydrophobic point of the partially denatured protein, acting as a molecular chaperone to prevent the further denaturing of the protein. In addition, the middle frag-

ment of dehydrin contains a large number of polar amino acid residues, which can produce synergistic effects with small polar molecules and low molecular weight substances (carbohydrates, amino acids, water molecules, etc.) in the nuclear matrix and cytoplasmic matrix, enhancing the protective effect of dehydrin on proteins [70].

### 3.3.3. Aquaporin

Aquaporin (AQP) is a class of intrinsic proteins in the plasma membrane or vacuolar membrane that specifically transport water, ranging from 26 kD to 30 kD, and belongs to the same family of major intrinsic protein (MIP) proteins as ion channels and glycerol channels. Based on the homology and structural characteristics of amino acid sequences, the AQPs family of plants is classified into four types: plasma membrane intrinsic proteins (PIPs); tonoplast intrinsic proteins (TIPs); nodulin 26-like intrinsic proteins (NIPs); small and basic intrinsic proteins (SIPs) [89]. Among them, PIPs are mainly located in the plasma membrane and can be divided into PIP1, PIP2, and PIP3 according to the homology difference between N-terminal and C-terminal sequences. TIPS are mainly distributed in the vacuolar membrane and can be divided into five groups according to different tissue location, namely $\alpha$, $\beta$, $\gamma$, $\delta$ and $\varepsilon$, which are important aquaporins in plants. NOD26 is the first member of the NIP family found in plants, located on the symbiotic membrane of soybean and rhizobia [90]. According to the structural differences of ar/R of aquaporins and the specificity of transport substrates, NIPS is divided into three categories: NIPI, NIPI, and NIPIII. This subfamily can transport other substances except for water molecules [91]. SIPs are the smallest family of AQPs in plants, mainly located in the endoplasmic omentum, and can be divided into SIP1 and SIP2 according to the different NPA sequences in the N-terminal and B-ring [92].

The expression of AQP showed strong temporal and spatial specificity. AQP is highly expressed in tissues and organs that need a lot of water flow, such as root epidermis, outer cortex and endodermis cells, xylem parenchyma cells near xylem vessels, phloem associated cells, guard cells, etc. The physiological function of AQP is closely related to its expression period and location, and its functions cover a series of physiological processes such as seed maturation and germination, cell elongation, root growth, leaf extension and movement, petal expansion, pollen, and ovule development [93–98]. At the subcellular level, AQP is mainly distributed in membrane systems such as cell membrane, vacuole membrane, endoplasmic omentum membrane, chloroplast membrane, and mitochondrial membrane. It was also found that AQP was redistributed at the subcellular level in different tissue sites and in different environments [99,100]. AQP located in the cell membrane at the cellular and subcellular levels is mainly responsible for water absorption and effluent. AQP located on the invaginated plasma membrane contributes to water transport between the protoplast and the vacuole. AQP located in the vacuolar membrane plays a role in regulating turgor. The specific distribution of plant AQP indicates that strong water flow across cells occurs in this region. In general, at the cellular level, the plasma membrane intrinsic protein (PIP) is mainly responsible for water absorption and outflow, and the vacuolar membrane intrinsic protein—tonoplast intrinsic protein (TIP)—is responsible for regulating turgor pressure, thus maintaining the integrity of cells [101]. For the whole plant, the specific distribution of plant AQP indicates that there is strong water flow across cells in this region [102].

AQP plays an important role in water transport. During the transmembrane transport of water in plants, AQP promotes the transmembrane transport of water inside and outside of cells by reducing the resistance encountered in the transmembrane transport of water and accelerates the rate of water migration between cells along the gradient of water potential. This is an important function of AQP in the transmembrane transport of water between different intracellular regions. At the same time, AQP is also the main way of water in and out of the cell, balancing the water potential inside and outside the cell. For example, the AQP on the cell membrane of plant root cells can regulate 70%~90% of the water flowing through the root. Water is absorbed by the root system

of the plant, which passes through the casparian strip into the vessels. The vascular system ensures that water is transported in large quantities through the plant. In many plants, AQP expression has been found in vascular bundles and adjacent tissues [103,104]. This suggests that plant AQP can accelerate water transport and facilitate water flow in and out of vascular bundles. In addition, plant AQP can maintain the water potential balance between xylem parenchyma cells and transpiration flow [105]. When the transpiration and water potential of the ducts are higher than that of parenchyma cells, the water will be stored in the vacuole through AQP transport. When the water potential of parenchyma cells is higher than the transpiration water potential of the ducts, AQP will transfer the stored water to the ducts. Water is transported across the plasma membrane and vacuole membrane of parenchyma cells through AQP. In addition to water molecules, aquaporin also transports other physiologically important neutral small molecules, such as $CO_2$, $H_2O_2$, glycerol, $NH_3/NH_4^+$, boron, silicon, and urea, which are involved in a series of important physiological processes in plants, such as photosynthesis, nutrient absorption, cell signal transduction, and stress response. The function of AQP determines its positive role in drought stress.

### 3.4. Reactive Oxygen Metabolism

3.4.1. Production and Basic Function of Reactive Oxygen Species

Oxygen is necessary for aerobic organisms to maintain their own life activities. When oxygen is not completely reduced in the metabolic process, a series of metabolites and their derivatives with more active chemical properties will be produced, called reactive oxygen species (ROS). ROS include superoxide radical $O_2^-$, $H_2O_2$, singlet oxygen $^1O_2$, hydroxyl radical $\cdot OH$, and organic oxygen radical ($RO\cdot$, $ROO\cdot$), etc. [106]. Under normal conditions, the ROS produced in plants maintains a balance with its scavenging system. However, when plants are under drought stress, ROS production and clearance will be out of balance. Drought can cause the increase of reactive oxygen free radicals and make plant cells suffer oxidative stress. When ROS exceeds the capacity of the ROS scavenging system, it will cause the accumulation of ROS and oxidative damage. The production of these free radicals will lead to a variety of harmful cytological effects, such as biofilm lipid peroxidation, protein denaturation, DNA strand breakage, and blocked photosynthesis. Two types of protection systems, enzymatic and non-enzymatic, have been formed correspondingly in the process of long-term evolution in plants to maintain a moderate level of ROS.

ROS can be produced in plants through many metabolic pathways. For example, in the process of photosynthesis and respiration, plant mitochondria, chloroplasts and peroxisomes, and some other organelles or parts with high oxidation activity or strong electron transfer function can also produce ROS. Chloroplasts are the main source of ROS production in green plants [107]. When plants are in a water-deficient environment, the absorption efficiency of light energy decreases. The blocked fixation of carbon dioxide in plants results in a decrease in $NADP^+$ supply and a relative increase in the rate of photosynthetic electron transfer to $O_2$. Oxygen and so on are used as electron acceptors to form $O_2^-$. In turn, $O_2^-$ can trigger a series of chain reactions to produce a large amount of ROS in plants [108]. Mitochondria are another important ROS-producing organelle. In the process of electron transfer in the respiratory chain, some electrons leak in the midway, making $O2$ form $O_2^-$ [109]. ROS in plants can also be produced in the plasma membrane and plasmid. NADPH oxidase, pH-dependent cell wall peroxidase, oxalate oxidase, and amine oxidase on the plasma membrane are all sources of ROS. Besides, enzymes in the endoplasmic reticulum and other organelles, such as cyclooxygenase, peroxidase, and lipoxygenase, can produce ROS through a series of chemical reactions.

The ROS function has two sides. ROS can destroy plant biofilm systems. For example, $\cdot OH$ can directly induce the peroxidation decomposition of the unsaturated fatty acid chain in phospholipids, thus destroying of membrane structure. However, the peroxides

and NO in ROS are mainly produced by NADPH oxidase, glutathione oxidase, and NO synthase, with low activity, so they cannot directly interact with lipids to induce lipid peroxidation (LPO). Pacher et al. showed that they can react quickly to produce peroxynitrite, which initiates the LPO reaction [110]. The forced destruction of membrane structure will lead to a series of biological dysfunction. ROS can also degrade biomacromolecules in plants. Almost all proteins or enzymes can be damaged by ROS oxidation. ROS can lead to decreased or loss of protein function, peptide chain breakage, protein crosslinking, the transformation of amino acid residues change, and changes in immunochemical properties, etc. [111]. The damage of ROS to protein is mainly through carbonylation and glycosylation. The oxygen-free radicals can interact with the sulfhydryl group of the active center of the enzyme to oxidize it into disulfide bonds, resulting in the inactivity of the enzyme. ROS can indirectly disrupt plant growth and development through the loss of enzyme activity. ROS can also interact with purines, pyrimidines, and deoxyribose in DNA molecules to cause the breakage, degradation, and modification of single or double strands of DNA, thus damaging genetic material [112]. In addition to the toxic effects of plant damage, ROS in plants is also involved in the process of resisting external stress and regulating plant growth and development. Oxidative burst is an important process in which ROS is involved in plant defense response. When the pathogen infects the plant, the plant produces a large amount of ROS through oxidative burst, which directly kills the pathogen [113]. In addition to biotic stress, ROS also plays a key regulatory role in response to abiotic stress [114].

### 3.4.2. Reactive Oxygen Scavenging System

In order to protect plants from ROS damage, there are endogenous antioxidant protection systems, including non-enzymatic antioxidants and antioxidant enzymes. The synergistic effect of antioxidants and antioxidant enzymes makes the production and quenching of ROS in vivo in a dynamic balance, thus alleviating or mitigating stress damage and making plants adapt to drought stress. The non-enzymatic scavenging systems of ROS in plants mainly include ascorbate, reduced glutathione (GSH), vitamin E, mannitol, carotenoids, and flavonoids. These substances can react directly with ROS or appear as substrates of enzymes in the ROS scavenging mechanism. In addition, some small molecules such as vitamins are also involved in scavenging oxygen free radicals and preventing lipid peroxidation. It is an indispensable part of the body's anti-oxidation defense system. Enzymes involved in antioxidant protection in plants mainly include SOD, CAT, APX, DHAR, MDHAR, GR, and POD. The main function of SOD is to remove $O_2^-$, and can convert $O_2^-$ to $H_2O_2$. SOD plays a key role in the enzyme system and is the first line of defense against ROS elimination system in plants. CAT and POD are mainly responsible for the removal of $H_2O_2$ in organisms. Besides, APX, GR, DHAR, and MDHAR are also very important $H_2O_2$ scavenging enzymes. Together, they form a second line of defense against ROS elimination systems in plants. GPX plays an important role in scavenging oxidative metabolism of lipids and alkyl peroxides, constituting the third line of defense against ROS scavenging.

SOD is one of the most important metal enzymes in the antioxidant enzyme system and plays a core role in the protective enzyme system. It alternately oxidizes and reduces the metals connected with the enzyme, and catalyzes the disproportionation reaction of $O_2^-$ to generate $O_2$ and $H_2O_2$. Its activity is considered to be an important index of plant stress resistance. Generally speaking, SOD activity in plants under drought stress is positively correlated with an antioxidant capacity [115]. SOD activity increased under mild or short-term water stress but decreased under severe or long-term water stress. However, some studies believe that the change of SOD activity is complex. For example, with the increase of stress intensity, SOD activity always decreases, or first decreases and then increases, or remains unchanged. The above differences may be due to the fact that the response of plants to water deficit is initiated not by water deficit itself, but by the degree of water deficit perceived by plants. Plant SOD can be divided into three types: Mn-SOD,

Cu/Zn-SOD, and Fe-SOD according to the metal atoms bound by SOD. Cu/Zn-SOD is composed of two subunits, each of which contains a Cu and a Zn, and is the most abundant one among the three superoxide dismutases. Each subunit of Mn-SOD and Fe-SOD contains only one metal ion. Mn-SOD and Fe-SOD have similar sequences and identical characteristic domains. Lower plants are dominated by Fe-SOD and Mn-SOD, while higher plants are dominated by Cu/Zn-SOD. Cu/Zn-SOD is mainly located in cytoplasm and chloroplasts, Mn-SOD is mainly located in mitochondria, and Fe-SOD is generally located in chloroplasts of some plants. In addition to cytoplasm, chloroplast, and mitochondria, SOD also exists in glyoxylate circulators and peroxisomes [116].

APx is one of the important components of the AsA-GSH redox pathway in plants. APx is about 30 kDa and generally exists in monomer form. Homodimer may also appear in some cAPx. It uses ascorbic acid (AsA) as an electron donor to catalyze the reaction between AsA and $H_2O_2$ to produce MD (monodehydroascorbate acid) and water. AsA, as both reactant and reaction product, can be recycled continuously, so that APx can be fully catalyzed to protect the chloroplast to maintain normal function. Four APx isozymes have been isolated: cytoplasmic isozyme cAPx, APx in chloroplasts, soluble sAPx in chloroplast stroma, and tAPx in membrane binding form in chloroplast thylakoids. In addition, a kind of peroxide object binding APx was also found. tAPx and sAPx exist in similar molar ratios in chloroplasts. Cytoplasmic cAPx and chloroplast APx have different electron donors and different internal sequences.

CAT is a heme-containing tetramer enzyme found in all plant cells that rapidly breaks down $H_2O_2$ into $H_2O$ and $O_2$. CAT mainly exists in peroxisomes in cells and is responsible for scavenging $H_2O_2$ produced in peroxisomes. CAT is also found in glyoxylic acid circulators and its function is mainly to remove $H_2O_2$ produced by photorespiration or fatty acid β-oxidation reaction [117]. Since $H_2O_2$ can be directly diffused across the membrane, $H_2O_2$ generated by other parts can also be diffused into peroxisomes and decomposed by CAT. In synergy with SOD, $H_2O_2$ can remove potentially harmful $O_2^-$ and $H_2O_2$ in plants, thus minimizing the formation of ·OH. CAT is not directly involved in the decomposition process of $H_2O_2$. Its scavenging mechanism is that the heme iron of the enzyme reacts with $H_2O_2$ to generate an iron peroxide active body, which then oxidizes 1 molecule of $H_2O_2$.

The non-enzymatic ROS scavenging system in plants mainly includes ascorbate, reduced glutathione (GSH), vitamin E, mannitol, carotenoids, and flavonoids, which can react directly with ROS or act as enzyme substrates in the ROS scavenging mechanism. In addition, as an indispensable part of the body's anti-oxidation defense system, some small molecules such as vitamins also participate in the removal of oxygen free radicals, preventing lipid peroxidation. For example, some cysteine-rich small molecular proteins in plants, such as metallothionein (MT) [118] and gibberellin-induced protein (GIP) [119], can also degrade $H_2O_2$. Overexpression of these antioxidant proteins can significantly reduce the content of $H_2O_2$ in plants after abiotic stress treatment, thus improving the stress resistance of transgenic plants.

Actually, there are two types of glutathione: reduced glutathione (GSH) and oxidized glutathione (GSSG). Among them, reduced glutathione (GSH) is commonly known as glutathione, which can scavenge free radicals in cells that have toxic effects. GSH is a mercapto tripeptide compound formed by the polymerization of glutamic acid, cysteine, and glycine, in which the mercapto group as the active group is easy to combine with some substances, such as free radicals and heavy metals to play a detoxification effect. In the biosynthesis of glutathione, GSH biosynthesis catalyzed by glutamate-cysteine ligase (GCL) and glutathione synthetase (GS) plays a crucial role in maintaining homeostasis and preventing redox damage [120]. For example, when a small amount of $H_2O_2$ is generated inside the cell, GSH reduces $H_2O_2$ to $H_2O$ under the action of GPx, and its own is oxidized to GSSG. Under the action of glutathione reductase, GSSG receives H to reduce to GSH, so that the scavenging reaction of free radicals in the body can be carried out continuously, thus stabilizing the membrane structure.

Ascorbic acid (AsA), also known as vitamin C, is a kind of abundant small molecule antioxidant substance commonly found in plants [121]. AsA can act as an important antioxidant and enzyme cofactor in plants, regulating photosynthesis, photooxidation, cell division, and playing an important role in plant signal transduction [122,123]. In plants, AsA content was positively correlated with plant stress resistance. The content of AsA varies greatly among different tissues of plants. For example, Smirnoff has suggested that AsA is present in chloroplast stroma in significantly higher concentrations than in other tissues [124]. As an important antioxidant in plants, AsA can directly or indirectly reduce the amount of ROS. AsA can directly remove ROS including $O_2^-$, $^1O_2$. Indirectly, AsA can reduce $\alpha$-tocopherol and act as an electron donor for APx to remove $H_2O_2$, thus achieving the ROS scavenging purpose [125]. In addition, AsA also plays an important role in photoprotection as a cofactor in the lutein cycle, thereby protecting organisms and their normal metabolism from damage caused by oxidative stress [126]. More importantly, because the end product of the AsA oxidation reaction is non-toxic DHA or 2, 3-DKG, the free radical reaction chain can be terminated.

The ascorbate-glutathione cycle (AsA-GSH) is the main pathway of AsA and GSH regeneration. In this cycle, AsA acts as an electron donor for ascorbate peroxidase (APx) to remove $H_2O_2$. Monodehydroascorbate (MDHA) generated by oxidation can be reduced by MDHAR, and can also disproportionate to generate AsA and dehydroascorbate (DHA). DHAR uses GSH as an electron donor to reduce DHA to AsA, and the oxidized glutathione (GSSG) generated can be reduced to GSH again by GR, so as to complete the process of scavenging ROS, such as $H_2O_2$ and regenerating AsA and GSH. The AsA-GSH cycle plays an important role in the antioxidant protection of plants under drought stress. A study on the response of AsA-GSH circulatory metabolic enzymes in *Coffea canephora* to drought stress showed that APX, GR, and DHAR activities increased under drought stress, but MDHAR activity had no significant change [127]. By studying sunflower and sorghum, Jingxian et al. found that there were differences in the response of AsA-GSH circulatory metabolic enzymes to drought stress in different plant organelles. They proposed that the chloroplast AsA-GSH cycle was the main method to remove $H_2O_2$ in sunflower under drought stress, while the cytoplasmic AsA-GSH cycle was the main method to remove $H_2O_2$ in sorghum [128].

## 4. Drought Stress Signal Transduction in Plants

The signal transduction process of plants from sensing environmental stimuli to responding to them generally includes three parts: (1) the sensory transduction and response of sensory cells to environmental stimuli, namely the original signal sensory transduction process, producing intercellular messenger; (2) the intercellular messenger is transmitted between cells or tissues, and finally acts on the receptor cell site; (3) the transduction and response of acceptor cells to intercellular messengers lead to physiological, biochemical, and functional changes in the acceptor tissues, which are ultimately reflected in the response of plants to environmental stimuli or adversity [129].

### 4.1. Plant Drought Stress Signal

The decrease of soil water content caused the change of leaf water status, and then affected the physiological function of plants. Leaf water potential reflects plant water status and is related to specific stress degrees. The decrease of leaf water potential and turgor pressure affected the synthesis, transportation, and distribution of plant hormones, such as ABA and cytokinin. Changes in turgor pressure caused by cell water loss may be the reason for cell perception of water stress, which is also known as the hydraulic signal of plant drought stress [130]. Besides hydraulic signal, the electrical signal also plays an important role in plant signal transduction under drought stress. Fromm et al. proposed through their study on maize in dry soil that electrical signals play an important role in the communication between roots and shoots of water-deficient plants [131]. What is more, when plants feel the initial drought signal, the osmotic stress signal is converted

into an intracellular chemical signal by the membrane receptor, which triggers the downstream effector to produce the second messenger. Then the signal is amplified gradually through the cascade transmission of the signal. In the process of signal transduction of dry early stress, the second messengers involved in signal transduction mainly included plant hormone signals, $Ca^{2+}$, IP3, phosphatidic acid, and ROS signals.

Plant hormones are a kind of chemical signal molecules that regulate plant growth. They often play a regulatory role in a low concentration. They can transmit cell signals in different parts of plants and among cells so that the remote transmission of plant signals can be realized. When soil water content decreases, some physiologically active substances act as chemical signals, and their content increases, which is called a positive signal. For example, under drought stress, the content of IAA, ABA, and ethylene increases. In contrast, a decrease in a biologically active substance is called a negative signal, such as cytokinins.

ABA is a small molecule lipophilic plant hormone, which is a crucial signal molecule in plant water stress. As a kind of plant hormone, ABA can control plant growth, inhibit seed germination and promote aging. In addition to regulating plant growth and development, ABA is also involved in regulating plant responses to various external stresses, embodied in content increasing greatly when the plant is in drought, high salt, low temperature, and other adversities. Moreover, ABA plays a pivotal role in the information connection between the aboveground and underground parts of plants. When plants are under drought stress, ABA produced in the rhizosphere can be used as a positive signal to regulate the physiological activities of aboveground parts. When plants are under water stress, root cells are the first to experience environmental changes and produce ABA, which transmits the signal to other organs and tissues of plants through vascular bundles, causing senescence of leaves and stomatal closure, so as to reduce water loss. ABA can be transported from the underground part to the aboveground part through the xylem, leading to increased ABA content in the leaves. In fact, ABA induces a wide range of downstream signaling factor responses, including kinases, phosphatases, G-proteins, and proteins in the ubiquitin pathway.

ABA has multiple receptors, such as ABAR/CHLH,•GCR2,•GTG1/2, and PYR/PYL/RCAR. These receptor proteins have the activity of protein kinases, which can be activated by binding ABA molecules to change the protein structure, and then activate or inhibit the activity of downstream signaling proteins to transmit signals between cells. Research on ABA receptors is still ongoing, and the exact function of the different receptors remains questionable. ABAR/CHLH is a magnesium ion chelatase H subunit located in plant cyto-plastids/chloroplasts. It not only catalyzes the synthesis of chlorophyll in cells but also participates in the reverse signal transfer between plastids/chloroplasts and the nucleus under stress conditions [132,133]. GCR2 protein is a G protein coupled receptor located in the plasma membrane of the cell. The C-terminal of GCR2 protein can interact with the A subunit of G protein (GPA1) to form a complex. The specific binding of ABA and GCR2 protein induces the release of G protein. The G protein is then separated into G$\alpha$ and G$\beta\gamma$ dimer, and the signal response of ABA is regulated by the downstream effector of GCR2 protein [134]. G protein, consisting of G$\alpha$, G$\beta$, and G$\gamma$ subunits, plays an important role in response to plant hormone signaling by synergistic G-protein coupled receptors and their downstream effectors. GTG1/2 was first identified and named by Pandey et al. through bioinformatics analysis. In the ABA signal transduction pathway model with GTG1/2 as the receptor, GPA1–GTP promoted GTG–GTP to maintain a high level by inhibiting GTG1/2 protease activity, thus reducing the binding probability of GTG–GDP and ABA. On the contrary, the binding of GTGSGDP to ABA can lead to the configuration change and then initiate ABA signaling response, but the specific molecular mechanism has not been clarified. The PYR/PYL/RCAR protein binds to ABA molecules outside the cell membrane, which in turn binds and inhibits the phosphatase activity of the downstream protein phosphatase PP2C [135].

As an essential mineral element in plants, $Ca^{2+}$ plays an important role in maintaining the stability of cell membrane and cell wall structure and participating in intracellular homeostasis and regulation of growth and development in terms of cell structure and physiological functions. Wang et al. found that extracellular $Ca^{2+}$ can activate the increase of intracellular $Ca^{2+}$ concentration through the calcium-sensing receptor (CAS) on the plasma membrane of guard cells of *Arabidopsis thaliana*, thus confirming the role of extracellular $Ca^{2+}$ as the first messenger [136]. In addition, as mentioned above, in response to drought, plants synthesize the hormone ABA, which causes stomatal closure to reduce water loss. During stomatal closure, the concentration of $Ca^{2+}$ in the cytoplasm increases, and $Ca^{2+}$ acts as the second messenger in osmotic stress response [137]. Drought-induced transient increase of intracellular $Ca^{2+}$ in guard cells promotes stomatal closure, maintains plant water, improves water use efficiency, and ultimately enhances plant adaptation to drought by interacting with or without ABA signaling pathways and downstream signal transduction mechanisms. In stomatal closure, the ABA-dependent $Ca^{2+}$ signaling pathway is the main pathway. ABA activates plasma membrane calcium channels in various ways and stimulates intracellular calcium reservoirs to release $Ca^{2+}$. More $Ca^{2+}$ will inhibit the inward potassium channel and further affect the anion channel. The phenomenon of anion outflow and depolarization will block the inward potassium channel and promote the outward potassium channel, leading to potassium ion outflow [138]. The guard cells are under low turgor pressure due to a large outflow of anions and potassium ions, making the stomata close gradually. IP3 and cyclic adenosine 5′-diphosphate ribose (cADPR) are also key second messengers in guard cells that can regulate $Ca^{2+}$ concentration. IP3 and cADPR can release $Ca^{2+}$ in guard cells and increase the concentration of $Ca^{2+}$, while ABA can rapidly increase IP3 and cADPR in guard cells. These three second messengers initiate calcium channels to transfer calcium ions into the cytoplasm and accumulate in large quantities, causing ion channels to interact with each other to produce a series of effects that promote stomatal closure [139,140]. $Ca^{2+}$ transmits stress signals downstream by interacting with protein receptors. Major $Ca^{2+}$ signal transduction pathways are involved in calcium-regulated kinase-mediated phosphorylation, including the regulation of downstream gene expression by $Ca^{2+}$ regulating transcription factors and $Ca^{2+}$ sensitive promoter elements [141]. Calcium-dependent protein kinases (CDPKs), calmodulin (CaM), and calcineurin B-like proteins (CBLs), which have been identified in plants, can recognize specific $Ca^{2+}$ and rely on these calcium signals to transmit downstream to adapt to drought stress.

A certain amount of ROS produced under stress can be used as signal molecules to activate relevant active substances or defense systems, and mitigate the damage caused by abiotic stress [142]. Among ROS, $H_2O_2$ is mostly used as an important signal molecule for animal and plant cells to respond to various stresses because $H_2O_2$ is a very stable ROS with the longest half-life and strong diffusivity. Different plant organelles have different responses to cellular REDOX signals under drought stress. Although $H_2O_2$ is produced faster in peroxisomes and chloroplasts, mitochondria are the most vulnerable organelles to oxidative damage [143,144]. Increased mitochondrial production of $H_2O_2$ may be an important alarm signal, up-regulating the antioxidant defense system or triggering programmed cell death when oxidative stress intensifies. Studies have shown that $H_2O_2$ can regulate calcium mobilization, protein phosphorylation, and gene expression. Pei et al. found that $H_2O_2$ can regulate $Ca^{2+}$ influx in protoplasts and increase of $[Ca^{2+}]cyt$ in guard cells by activating $Ca^{2+}$ channels in the plasma membrane of guard cells of *Arabidopsis thaliana*. In addition, they further proposed that ABA-induced $H_2O_2$ production and $H_2O_2^-$activated $Ca^{2+}$ channels are important mechanisms of ABA-induced stomatal closure [145]. Mori et al. also reported an inevitable link between ROS signaling and stomatal closure in plants [146]. Yan et al. also reached the same conclusion: ABA can promote the production of ROS, and the ROS produced can act as signal molecules to regulate stomatal closure [147]. In addition, $H_2O_2$ also induces the phosphorylation of mitogen-activated

protein kinase (MAPK), which is involved in multiple signal transduction cascades that regulate downstream gene expression [148].

*4.2. Intracellular Transduction Pathways and Regulation Mechanisms of Plant Drought Stress Signals*

Drought stress signal transduction can be divided into two pathways. The first pathway is the ROS-activated MAPK cascade pathway. MAPK cascade regulates antioxidant defense system and osmotic regulation system in plants. Furthermore, the damage caused by drought stress can be relieved by removing ROS and changing the osmotic potential of cells. The other pathway is $Ca^{2+}$-dependent stress signaling bypass mediated by calmodulin-dependent protein kinase (CDPK). $Ca^{2+}$ signal is produced under drought stress, and $Ca^{2+}$ signal further regulates the expression of plant protective proteins, such as LEA protein through CDPK, which is involved in the late response to drought stress, and ultimately enhances the drought resistance of plants.

Mitogen-activated protein kinases (MAPKs) are a class of important protein kinases involved in signal transduction, which play an extremely important role in plant growth, development, and stress response [149]. The MAPK cascade consists of three components: MAPK, MAPKK (MAPK kinase), and MAPKKK (MAPK kinase kinase). When the first member of this pathway, MAPKKK, is activated, the other two components undergo sequential phosphorylation and are activated in turn. The reason is that MAPKKK can double phosphorylate the serine (Ser) and serine/threonine (Ser/Thr) in MAPKK, thus activating it. The protein kinase of MAPK containing *n* conservative district and a very conservative TXY motif between the VII and the III subregion [150]. MAPKK initiates MAPKK by dual phosphorylation of threonine (T) and tyrosine (Y) residues at both ends of the X site [151]. As a result, MAPKK phosphorylates MAPKK and MAPKK phosphorylates MAPKK. Activated MAPKK can activate transcription factors and also cause cellular signaling responses through interactions with other proteins.

The full name of CDPK is calmodulin-dependent/calmodulin-independent protein kinase or calmodulin-like domain protein kinase. It belongs to Ser/Thr type protein kinases and is a large family encoded by multiple genes. Under the stimulation of external signals, plant cells showed changes in $Ca^{2+}$ concentration and then activated CDPK. CDPK regulates downstream gene expression and product activity through the phosphorylation cascade. These products play an important role in the regulation of gene expression, enzyme metabolism, ion, and water transmembrane transport, and other microscopic aspects so that plants show macroscopic changes such as growth and development, stress resistance changes [152].

## 5. Drought Stress Signal Transduction in Plants

Generally, drought stress response genes can be divided into functional genes and regulatory genes. The products of functional genes directly resist environmental stress, such as aquaporin genes, osmoregulatory factors (such as sucrose, proline, and betaine) synthase genes, protective proteins (such as LEA protein, molecular chaperone, etc.) genes. The products of regulatory genes, such as protein kinase genes, protein phosphatase genes, phospholipid metabolism-related genes, and stress-related transcription factor genes, are involved in signal transduction and regulation of gene expression to indirectly respond to stress. These proteins act by participating in plant stress signal transduction pathways or by regulating the expression and activity of other effector molecules.

*5.1. Functional Genes*

5.1.1. Osmotic Adjustment Related Genes

According to the different pathways of proline accumulation, the related enzymes can be divided into three categories. The first category is the enzymes related to proline

synthesis, including △-pyrroline-5-carboxylate synthetase (P5CS), pyrroline-5-carboxylate reductase (P5CR), and ornithine-δ-aminotransferase (δ-OAT). The second category is related to the degradation of proline enzymes, including proline dehydrogenase (ProDH) and △-pyrroline-5-carboxylate dehydrogenase (P5CDH). The third category is proline transport-related enzyme ProT. The synthesis sites of proline in plants are cytoplasm and chloroplast, and the synthesis pathways include glutamic acid (Glu) and ornithine (Orn) synthesis pathways [153]. Glutamic acid synthesis pathway mainly occurred under osmotic stress and nitrogen deficiency, while ornithine synthesis pathway existed in nitrogen abundant environment [154]. In the glutamic acid synthesis pathway, Glu is catalyzed by △-pyrroline-5-carboxylate synthetase (P5CS) to produce glutamic semialdehyde (GSA). Subsequently, GSA is automatically cycled to form pyrroline-5-carboxylic acid (P5C), which generates proline (Pro) under the action of pyrroline-5-carboxylate reductase (P5CR) [155,156]. Substrates and enzymes in the first step of the ornithine synthesis pathway are different from those in the glutamate pathway. The substrate was ornithine (Orn) and the enzyme was ornithine-δ-aminotransferase (δ-OAT). The substrates and products under the two pathways mainly include Glu, Orn, GSA, P5C, and Pro. The enzymes required for the reaction include P5CS, P5CR, and δ-OAT. Kishor et al. transferred the *P5CS* and *P5CR* genes into tobacco. It was found that although the mRNA levels of both were increased, the proline level of *P5CR* transgenic tobacco was not significantly increased, while the proline level of *P5CS* transgenic tobacco was significantly increased [157]. La Rosa et al. obtained the same result that when soybean *P5CR* gene was overexpressed in tobacco, the activity of P5CR was increased five times, but the level of proline in transgenic tobacco was not significantly increased [158]. These results indicated that the increase of proline was more affected by *P5CS* than by *P5CR*. Therefore, the P5CS enzyme is the rate-limiting enzyme of proline metabolism and determines the synthesis of proline. Sharma et al. found that Arabidopsis *P5CS1* mutants underproduce proline during stress [159]. Baocheng et al. introduced P5CS cDNA from moth bean (*Vigna aconitifolia* L.) into rice (*Oryza sativa* L.) genome. The transgenic plants showed overproduction of the P5CS enzyme and accumulation of proline [160]. Similarly, in transgenic *AtP5CS* tobacco, its proline content was significantly increased, and its osmotic regulation ability was enhanced [161]. The same effect was also shown in potato [162], sugarcane [163], soybean [164], etc. δ-OAT is another key enzyme in proline synthesis and its activity was significantly enhanced under drought conditions [165]. Overexpression of *δ-OAT* in plants can significantly increase proline content in tobacco, rice, etc. [166,167]. In addition, the degradation of proline occurs in mitochondria and is the reversal of the synthesis pathway of glutamic acid. Proline is first oxidized by proline dehydrogenase (ProDH) to P5C, which is reduced to glutamic acid by △-pyrroline-5-carboxylate dehydrogenase (P5CDH) [168]. Studies have shown that Arabidopsis proline dehydrogenase (*PDH1*) mutants block Pro catabolism and found that plants maintain growth through active Pro catabolism under low water potential [159]. What is more, proline transport requires the participation of ProT. This transporter belongs to the amino acid/auxin permease (AAAP) gene family in plants and is a typical $Na^+$-dependent sub-amino acid transporter. The transporter is directly absorbed by proline coupling along with the $Na^+$-electrochemical gradient, which requires the participation of $Na^+$-K-ATPase and belongs to active transport [169]. However, many studies have proved that the alteration of ProT expression cannot change proline accumulation in a directed way. In *Arabidopsis thaliana* plants overexpressing *HvProT*, the proline content in the aboveground part decreased while that in the root increased [170].

The synthesis of glycine betaine (GB) in plants, mainly accomplished by the enzymatic reaction, has been elucidated in many studies. Choline, as the initiator of GB synthesis, is obtained through the methylation of three adenosine-methionine-dependent phospho-ethanolamine (PE) catalyzed by the cytoplasmic enzyme phospho-ethanolamine N-methyltransferase (PEAMT) [171]. The PEAMT enzyme has two tandem methyltrans-

ferase domains at the N terminal and C terminal. The N-terminal methyltransferase domain methylate PE to phosphate-monomethyl-ethanolamine (P-MME), and the C-terminal methyltransferase domain methylate P-MME to phosphate-dimethylethanolamine (P-DME), and P-DME to phosphocholine (PC) [172]. PC is then converted to choline in different ways. McNeil et al. found a different transformation pathway for PC in spinach and tobacco, the former by direct dephosphorylation to choline, and the latter by first containing PC in phosphatidylcholine and then metabolizing it to choline [173]. Next, betaine is synthesized by a two-step oxidation reaction. The first step was to oxygenate choline into betaine aldehyde with the help of a ferredoxin-dependent choline monooxygenase (CMO). The CMO catalyzed step is the rate-limiting step in GB biosynthesis [174]. The second step is $NAD^+$-dependent betaine aldehyde dehydrogenase (BADH) catalyzed the oxidation of betaine aldehyde into betaine [175,176]. CMO is a ferredoxin-dependent rate-limiting enzyme encoded by a single gene. CMO has Rieske-type [2Fe-2s] active site and is the only matrix enzyme with the Rieske iron-sulfur center, usually localized in the chloroplast or other subcellular compartments [177]. Under normal conditions, CMO activity is low and unstable. Since the reduced ferredoxin is produced by photosynthetic electron transport, the CMO activity in plants can be improved to a certain extent under light induction. The CMO plays a balancing and speed-limiting role in this process. Due to the toxic effect of betaine aldehyde on plant cells in this step, CMO should not only synthesize enough betaine aldehyde for further synthesis of betaine but also limit the excessive accumulation of betaine aldehyde in the plant. The catalytic enzyme BADH is a dimer encoded by a single chain nuclear gene with two alleles. It is composed of two monomers of equal molecular weight. It belongs to the superfamily of aldehyde dehydrogenases and also has nonspecific effects on other aldehyde substrates [178]. BADH is dependent on both $NAD^+$ and $NADP^+$, but in plants, BADH shows higher activity in the presence of $NAD^+$ [179]. The BADH of monocotyledons may be located in microsomes, while that of dicotyledons may be located in the chloroplast stroma. BADH has two isozymes (BADH I and BADH II), in which BADHII plays a more important role [180]. BADH, as the most important catalytic enzyme in the synthesis of betaine, has low activity under normal conditions. However, under the stress conditions of low temperature, drought, and high salinity, the isozyme activity of BADH was significantly increased, which resulted in the synthesis of a large amount of betaine, indicating that the activity of BADH was induced by stress. With advances in genomics and proteomics as well as genetic engineering techniques, some plant species have been engineered using genes from the GB biosynthetic pathway that confer tolerance to abiotic stresses. Most of the plants that have been genetically engineered to produce GB are naturally non-GB accumulative plants [181]. Shen et al. isolated and identified the *CMO* gene from spinach and transferred it into tobacco, and found that salt tolerance and drought tolerance of transgenic tobacco were also significantly improved [182]. Similarly, other studies have also shown that *CMO* transgenic rice and tobacco can significantly improve their tolerance to salt and drought stress [183,184]. Ishitani et al. isolated and cloned the *BADH* gene from barley and transferred it into tobacco, which improved the drought tolerance of tobacco to a certain extent [185]. Fan et al. transferred the *SoBADH* gene from spinach into the sweet potato and found that the transgenic plants showed stronger BADH activity and eventually showed increased tolerance to abiotic stress [186]. Li et al. transferred the *SoBADH* gene into tomatoes to produce transgenic plants with higher levels of betaine and greater stress resistance [187].

The metabolism of soluble sugar in plants is very complex. Taking sucrose as an example, FBPase (fructose-1, 6-bisphosphatase) and sucrose phosphate synthase (SPS) are important rate-limiting enzymes in the sucrose synthesis pathway. The enzyme FBPase, one of the key enzymes in the gluconeogenesis pathway, catalyzes the hydrolysis of fructose-1 6- diphosphate (FDP) to fructose -6- phosphate (F6P). The catalytic product of FBPase in the cytoplasm is sucrose, while the catalytic product of FBPase in the chloroplast is starch. Cho et al. constructed FBPase overexpressed Arabidopsis lines and found that the soluble sugar content of the transgenic plants was significantly increased [188]. On the

contrary, decreasing the activity of FBPase in potato cytoplasm by antisense technique resulted in a decrease in sucrose synthesis rate [189]. SPS catalyzed the synthesis of sucrose-6-phosphate using uridine diphosphate glucose (UDPG) as the donor and fructose 6-phosphate as the receptor. Sucrose 6-phosphate is dephosphorylated and hydrolyzed to form sucrose and phosphate ions under the action of sucrose phosphate phosphorylase (SPP). This reaction is basically irreversible. However, SPS and SPP exist in the plant body in the form of complex, so SPS catalysis of sucrose production is actually irreversible. Therefore, SPS is a key enzyme controlling sucrose synthesis in plants [190]. Park et al. transferred the Arabidopsis *AtSPS1* gene into tobacco and found that the sucrose content of transgenic tobacco increased, accompanied by plant height growth, stem diameter thickening, and fiber lengthening [191]. Moreover, previous studies have confirmed that the SPS activity and sucrose content of transgenic plants obtained by introducing *ZmSPS1* into tomato [192], potato [193], and Arabidopsis [194] were significantly increased.

5.1.2. Drought-Induced Protein Genes

LEA protein is a protein that is highly expressed in late embryonic development. It plays a crucial role in plant response and resistance to drought, mainly by capturing water, stabilizing and protecting the structure and function of proteins and membranes, and protecting cells from water stress as a molecular chaperone and hydrophilic solute [195]. Sivamani introduced the ABA-responsive gene *HVA1* (a member of group 3 LEA protein genes) into spring wheat (*Triticum aestivum* L.), and found that the transgenic wheat had significantly higher water use efficiency and better growth characteristics under water deficit condition than the control wheat [196]. Under drought stress, seed germination rate, seedling fresh weight, and root length of *CmLEA-S* (a melon Y3SK2-type *LEA* gene) transgenic plants were significantly higher than those of wild-type plants. They also had less wilting and yellowing, more proline, less MDA, and stronger APX and CAT activities [197]. Luo et al. constructed *Capsicum annuum* L. plants with the expression of *CaDHN5* (a dehydrin gene) downregulated by VIGS (Virus-induced Gene Silencing) and Arabidopsis plants with transgenic overexpression of *CaDHN5*. It was found that *CaDHN5* was positively correlated with the expression of manganese superoxide dismutase (*MnSOD*) and peroxidase (*POD*) genes [198]. Under drought stress, seed germination rate and survival rate of *OeSRC1* (a Ks-type dehydrin gene) transgenic tobacco plants were higher than those of wild-type tobacco plants, and they accumulated more free proline, but electrolyte leakage did not change significantly [199].

Plant aquaporin (AQP) is a membrane channel located in the plasma membrane and intracellular module, which can promote the transport of water, small neutral molecules, and gases across biofilm [200]. Aquaporin belongs to the MIP family of proteins that regulate cellular water movement and maintain water relationships in plants, especially under drought stress. As mentioned earlier, AQP can be divided into PIPs, TIPs, NIPs, and SIPs, as well as the genes that encode them. Among them, plasma membrane intrinsic proteins (PIPs) and tonoplast intrinsic proteins (TIPs) mediate the main pathways of intracellular water transport, maintain intracellular and intercellular water relations under stress, and are involved in many processes of the drought stress response. Zhang etc. found that rose water channel protein RhPIP2;1 can influence plant growth and stress reaction by interacting with the membrane MYB protein RhPTM [201]. Overexpression of *CrPIP2;3* in *Arabidopsis thaliana* (a PIP2 gene from rose) can promote the survival and recovery of transgenic plants under drought stress by regulating water homeostasis, thus affecting drought tolerance of plants [202]. The seed germination rate, seed yield, seed vigor, and root length of transgenic *Arabidopsis thaliana* lines overexpressing *JcPIP2;7* (a plasma membrane intrinsic protein gene) and *JcTIP1;3* (a tonoplast intrinsic protein gene) under mannitol condition were significantly higher than those of the control [203]. Peng et al. tested the effect of the ginseng *PgTIP1* gene by transgenic it into Arabidopsis plants and showed that it altered root morphology and leaf water channel activity, thereby al-

tering drought tolerance [204]. The overexpression of *CsTIP2;1* in Arabidopsis plants increased the expansion of mesophyll cells, midrib aquiferous parenchyma abundance, $H_2O_2$ detoxification, and stomatal conductance, and then significantly improved the water and oxidation state, photosynthetic capacity, transpiration rate, and water use efficiency of leaves under the condition of continuous dry soil [205].

### 5.2. Regulatory Genes

Regulatory genes are genes that regulate stress signal transduction and functional gene expression. The regulatory genes of drought stress response can be divided into the following categories. The first is the transcription factors related to the regulation of stress gene expression, including bZIP, MYB, MYC, EREBP/APZ, CBFI (CRT/DRE binding factor), DREB1A (DRE binding), etc. These transcription factors can be strongly induced by water stress and their expression can further regulate the expression of various functional genes. The second type of protein kinases is related to the sensing and transduction of stress signals, such as receptor protein kinases, ribosomal protein kinases, transcription regulatory protein kinases, etc. These kinases usually play the role of stress signal cascading amplification. Among them, the most important are the three key kinases included in the MAPK cascade: MAPK, MAPKK, and MAPKKK. The third type is related to the second messenger generation and transduction of enzymes, such as phospholipase D, phospholipase C. Phospholipase C catalyzes the hydrolysis of PIP into diesterphthalein glycerol (DG) and inositol triphosphate (IP3). IP3 can induce the release of $Ca^{2+}$ stored in the endoplasmic reticulum into the cytoplasm, and thus initiate the intracellular signal transduction process.

### 5.2.1. Signal Transduction Related Genes

The key step of ABA biosynthesis is to catalyze 9-cis-epoxycarotenoid dioxygenase (NCED) [206]. In *Arabidopsis thaliana*, drought tolerance is regulated by the *NCED* gene. The overexpression of the *AtNCED3* gene in Arabidopsis leads to the increase of endogenous ABA level, and drought and ABA promote gene transcription. Overexpression of this gene in plants resulted in a decrease in leaf respiration rate and an increase in drought resistance. The antisense inhibition of this gene made it sensitive to drought, suggesting that the expression of this gene plays a key role in ABA biosynthesis under drought stress [207]. Under drought stress, the increased activity of ABA synthase (such as *ZEP*, *NCED*, *LOS5/ABA3*, and *AAO*) in plant root cells produced a large amount of ABA, which was transported to leaf cells through transpiration flow. ABA is perceived by ABA receptors on guard cells and is transported across the membrane by intracellular second messengers [calcium messenger, proton messenger, inositol triphosphate (IP3), etc.]. Thus, a variety of ion channels and enzymes related to physiological and biochemical reactions are activated to regulate stomatal movement and eventually lead to stomatal closure. Other studies have shown that under drought conditions, ABA promotes open stomatal closure and inhibits closed stomatal opening in isolation. During stomatal closure, ABA, $H_2O_2$, and NO may all act on the MAPK signaling pathway. In the future, tomato-derived *LeNCED1* was transferred into tobacco (with tetracycline as control). When tobacco leaves were treated with tetracycline, the increase of ABA content in the leaves induced *NCED* transcription, but there was no significant difference in the tomato transformed with *LeNCED1* under the strong promoter CaMV35S [208].

Calmodulin, calmodulin-like proteins, calmodulin B-like proteins, and calcium-dependent protein kinases (CDPKs) are the four major families of calcium-binding proteins in plants. As a $Ca^{2+}$ signal sensor, CDPK is closely related to the further transmission of cellular $Ca^{2+}$ signal. Because the N-terminal serine/threonine-protein kinase domain of CDPKs can be fused with the carboxy-terminal calmodulin-like domain containing the EF-hand calcium-binding site, CDPKs are independent of exogenous calmodulin interactions but can be directly activated by $Ca^{2+}$ binding [209,210]. Although most *CDPK* genes

are commonly expressed in organisms, some *CDPK* genes are expressed only in specific tissues or are induced by hormonal, biological, or abiotic stress conditions. Salt stress, drought stress, and other abiotic stress can significantly improve the transcription level of CDPK [211,212]. Urao et al. cloned two *CDPK* genes, named *AtCDPK1* and *AtCDPK2*, from *Arabidopsis thaliana*. The expression of these two genes can be induced by drought, suggesting that these two genes are involved in osmotic stress signal transduction [213]. The protein kinases *AtCDPK10* and *AtCDPK30* expressed in maize protoplasts can activate the promoter of the *HVA1* gene induced by drought and high salt stress, and the mutant without the *CDPK* region is not responsive to various stresses and ABA. Therefore, it is speculated that *AtCDPK10* and *AtCDPK30* are the positive regulators of the plant stress signal transduction pathway [214]. Moreover, Saijo et al. found that overexpression of *OsCDPK7* in rice enhanced drought stress resistance of rice [215].

The MAP protein kinase genes isolated from *Arabidopsis thaliana* were induced by drought, high salinity, and low-temperature stress, including *AtMPK3*, *AtMPK4*, *AtMPK6*, *AtMEK1*, and *AtMEKK1*. Studies have shown that the MAP kinase cascade system is not only regulated by phosphorylation and dephosphorylation at the protein level but also induced by environmental stress signals at the transcriptional level. Mizoguchi et al. found that *AtMEKK1* is involved in the MAP kinase cascade signaling of drought, high salinity, low temperature, and traumatic stress in Arabidopsis. The cascade pathway consists of AtMEKK1 (MAPKK kinase), AtMEK1 (MAPKK kinase), and AtMPK4 (MAPKK kinase) [216]. It has also been reported that drought or high salinity also activates SIMK (stress-induced MAPK) in *Medicago sativa* cells and SIPK (salicylic acid-induced protein kinase) in tobacco cells. Chitlaru et al. found that hypertonic stress could rapidly activate a protein kinase, and confirmed that the protein kinase belonged to MEK1 [217]. Xiong et al. found that the *OsMAPK5* gene in rice was induced by a variety of biological and abiotic stresses, and overexpression of this gene in rice could enhance drought resistance, salt resistance and low-temperature tolerance of transgenic rice [218].

5.2.2. Transcription Factor Genes

In the process of signal transduction under drought stress, transcription factors (TF) regulate and reduce the damage to plants from multiple levels by activating multiple pathways, which plays a crucial role in the growth and development of plants under stress [219]. Among them, the transcription factor gene families related to drought stress mainly include *HD-Zip/bZIP*, *AP2/ERF*, *NAC*, *MYB*, and *WRKY*. However, Different transcriptional factors play different transcriptional regulatory roles under drought conditions, depending on plant species and strain, development stage, and drought treatment intensity. Gong et al. pointed out that the 43 transcription factor genes in drought response of tomato drought-resistant lines mainly came from 5 families with the most abundant expression changes, which were *WRKY*, *NAC*, *BHLH*, *AP2/EREBP*, and *HSF* in turn, while *MYB*, *bZIP*, and *CCAAT* families had less abundant expression [220]. Different from tomato, the highest abundance of the 261 transcription factor genes in rice were *MYB* (35 members) and *AP2/EREBP* (28 members), followed by 21 *bHLH*, 11 *HSF*, 27 *NAC*, and 15 *WRKY*. Moreover, drought-resistant rice cultivars could activate more upregulated transcription factors than non-drought-resistant rice cultivars. For example, the number of upregulated transcription factors in the *AP2* family of rice drought-resistant variety was 35 more than that of rice non-drought-resistant variety after 18 days of drought [221].

The HD-Zip transcription factors belong to a homeobox protein encoding 60 conserved amino acid homeodomains (HD), which consists of six families, namely HD-Zip, KNOX, PHD, BELL, WOX, and ZF-HD [222]. Among them, homeodomain-leucine zipper (HD-zip) is a plant-specific transcription factor, which consists of DNA-homologous domain and additional Leu zipper (Zip) components [223]. The former binds specifically to DNA, while the latter mediates the formation of protein dimer, a transcription factor involved in regulating plant growth and development under normal growth conditions and environmental stress [224]. Based on sequence conservatism, structural characteristics,

function, and other characteristics, HD-Zip transcription factors can be divided into four subfamilies (HD-Zip I~ HD-Zip IV). Different subfamily members have different biological functions, some are involved in the cross-interaction of multiple hormonal pathways, and some interact with key genes and downstream genes of hormonal pathways [222]. Atalou et al. proposed that the expression of subfamily I and II genes of the HD-Zip family of transcription factors were induced by drought stress. These two genes participate in the hormone signaling pathway, regulate the expansion, division, and differentiation of plant cells by interacting with the hormone pathway genes and downstream genes, and thus improve the drought resistance of plants [225]. Expression analysis by Deng et al. showed that CpHB-7 negatively regulates the expression of ABA-responsive genes, which also explains the reduced sensitivity of transgenic plants with ectopic *CpHB-7* to ABA during seed germination and stomatal closure [226]. *Arabidopsis thaliana* with overexpression of HD-Zip I subfamily gene Hahb-4 showed strong tolerance to water stress and insensitivity to ethylene because the overexpression of *Hahb-4* gene inhibited the expression of ethylene synthesis genes *ACO*, *SAM*, and downstream ethylene signaling genes *ERF2* and *ERF5* [227]. Fan et al. silenced RhHB1, which encodes a homeodomain-leucine zipper I γ-clade transcription factor in rose flowers, resulting in an increased content of JA-Ile and a decreased tolerance to dehydration. It has also been shown that RhHB1 can inhibit the expression of lipoxygenase 4 (*RhLOX4*) by directly binding to the promoter of *RhLOX4*. In other words, the JA feedback loop mediated by the *RhHB1/RhLOX4* regulatory module provides dehydration tolerance by fine-tuning the level of bioactive JA [228].

AP2/ERF transcription factors play an important role in plant stress resistance and previous studies have shown that they can participate in the process of drought stress resistance in plants through different pathways. AP2/ERF can regulate drought stress response by affecting the synthesis of plant hormones. Cheng et al. proposed that as the upstream component of jasmonic acid and ethylene signals, ERF1 can integrate JA, ET, and abscisic acid signals through stress-specific gene regulation, and play a positive role in drought tolerance [229]. Wan et al. found a drought-induced upregulated ERF transcription factor gene *OsDERF1*, and the overexpression of *OsDERF1* in rice reduced the tolerance of rice to drought stress at the seedling stage. It has been demonstrated that *Os*DERF1 can directly bind to the GCC boxes in the promoter regions of negative regulatory factors *OsAP2-39* and *OsERF3* and activate their expression. However, *Os*AP2-39 and *Os*ERF3 can bind to the GCC box of *ACS* and *ACO* promoter of ethylene synthesis genes and inhibit the expression of these genes, thus inhibiting the synthesis of ethylene. Therefore, the reduction of ethylene content by overexpression of *OsDERF1* is one of the important reasons for the decrease of drought tolerance in rice [230]. Zhang et al. found that overexpression of *JERF1* can improve drought tolerance of transgenic rice and that *JERF1* can activate expression of *OsABA2* and *Os03G0810800*, two key enzymes of ABA synthesis, and increase ABA content. These results suggest that *JERF1* may regulate drought response through the ABA pathway. Moreover, AP2/EREBP can also respond to drought stress by affecting metabolite synthesis in plants. By overexpressing *DREB1A* in *Arabidopsis thaliana*, Maruyama et al. found that the contents of starch degrading enzyme, sucrose metabolizing enzyme, and sugar alcohol synthase changed, affecting the content changes of monosaccharide, disaccharide, and trisaccharide, thus enhancing the drought resistance of transgenic plants [231]. In upland rice, *OsERF71*-overexpressing lines, different from *OsERF71* interference lines, were found to enhance drought resistance by increasing the expression of *OsP5CS1* and *OsP5CS2* regulating proline synthesis [232]. The overexpression of *FaDREB2* in *Broussonetia papyrifera* can increase the content of soluble sugar and proline in vivo, and thus enhance the tolerance [233]. It was also found in rice that when the rice gene *JERF3* was overexpressed, the accumulation of sugar and proline in rice could be increased to resist drought [234]. Similarly, overexpression of *GmERF3* could improve the drought resistance of tobacco by increasing soluble sugar and proline content, respectively [235]. In addition, some people have pointed out that AP2/EREBP pro-

tein is also involved in ROS clearance. Through GUS activity test and SOD activity detection, Wu et al. found that JERF3 can bind to the GCC box of *NtSOD*, thereby activating the expression of *NtSOD*, improving the activity of SOD, enhancing the ability of ROS scavenging, and improving the tolerance of tobacco to osmotic stress [236]. What is more, AP2/ERF transcription factor can also regulate drought resistance of plants by participating in the regulation of wax synthesis. Wang et al. found that OsWR1 physically interacts with the DRE and GCC boxes in the promoter of wax-related genes *OsLACS2* and *OsFAE1′-L*, which can directly regulate the expression of these genes, thereby altering long-chain fatty acids and alkanes to regulate wax synthesis. Therefore, the drought resistance of overexpression of *OsWR1* was significantly improved [237].

MYB is one of the largest transcription factor families in plants. It is widely involved in the regulation of secondary metabolism, response to hormones and environment, the guidance of cell differentiation and morphogenesis, and also plays a key role in resistance to drought and other abiotic stresses [238]. The N-terminal of MYB transcription factor is a conserved helix-turn-helix (HTH) protein DNA binding domain consisting of 52 amino acids, which directly determines the accuracy of binding to target genes and can bind to cis components, such as GCC box, DRE, ABRE, W box, etc. The C-terminal is the transcriptional initiating region, which determines the transcriptional activity of a transcription factor and its interaction with other genes or components to manipulate the expression efficiency of downstream genes [239]. According to the structure of the DNA binding domain, the MYB transcription factor family can be divided into 1R-MYB, 2R-MYB, 3R-MYB, and 4R-MYB subfamilies. In Arabidopsis plants overexpressing *OsMYB3R-2*, expression of genes dehydration-responsive element-binding protein 2A, *COR15a*, and *RCI2A* was significantly increased, leading to enhanced abiotic stress resistance [240]. The *Arabidopsis thaliana* overexpressing *GaMYB85* had higher free proline and chlorophyll content, showed higher seed germination rate under mannitol treatment, and higher drought resistance efficiency than the wild type under water shortage conditions, most probably via an ABA-induced pathway. Furthermore, the ectopic expression of *GaMYB85* resulted in increased transcription levels of stress-related markers such as *RD22*, *ADH1*, *RD29A*, *P5CS*, and *ABI5* [241]. *GbMYB5* gene silencing decreased the proline content and antioxidant enzyme activity increased the malondialdehyde (MDA) content and decreased the tolerance of cotton to drought stress. However, in tobacco lines overexpressing *GbMYB5*, proline content and antioxidant enzyme activity increased, while MDA content decreased. The expression levels of the antioxidant genes *SOD*, *CAT*, and *GST*, polyamine biosynthesis genes *ADC1* and *SAMDC*, and the late embryogenesis abundant protein-encoding gene ERD10D and dry-responsive genes *NCED3*, *BG*, and *RD26* were significantly increased in tobacco overexpressing *GbMYB5* [242]. Wang et al. constructed *GmMYB84* overexpressing soybeans, which has longer primary root length, greater proline, and ROS contents, higher antioxidant enzyme activities [peroxidase (POD), catalase (CAT), and superoxide dismutase (SOD)], lower dehydration rate, and reduced malondialdehyde (MDA) content. In addition, they found that some ROS-related genes of the transgenic plants were upregulated under abiotic stress, and *GmMYB84* could directly bind to the promoters of *GmBOHB-1* and *GmBOHB-2* genes through electrophoretic mobility shift assay and luciferase reporter analysis [243]. Chen found that MdMYB46 can directly bind to lignin biosynthesis-related gene promoter to promote secondary cell wall biosynthesis and lignin deposition, and can also directly activate stress response signals to improve salt and osmotic stress tolerance of apple [244]. Geng also found that MdMYB88 and MdMYB124 could regulate root xylem development and regulate cellulose and lignin accumulation in response to drought by directly binding to *MdDVND6* and *MdMYB46* promoter under drought conditions [245].

The WRKY protein family, named for its highly conserved WRKYGQK DNA domain, is a zinc finger-type transcription regulator, which is a unique transcription factor in plants. In addition to the presence of at least one highly conserved WRKYGQK sequence and zinc finger structure, the WRKY domain also specifically interacts with the (T)

(T) TGAC (C/T) sequence (W box) of the target gene promoter [246]. W-boxes are found in the promoters of many genes related to plant defense response and even in the self-promoters of some WRKY transcription factor genes. Therefore, WRKY transcription factors may regulate the expression of downstream functional genes or other regulatory genes through binding with W-box, thus participating in the regulation process of various physiological activities in plants. Overexpression of *TaWRKY10* in tobacco enhanced drought resistance, which was characterized by higher proline and soluble sugar content, lower ROS, and MDA content, and increased germination rate, root length, survival rate, and relative water content under stress conditions. This is because TaWRKY10 plays a positive role in drought stress by regulating osmotic balance, scavenging ROS and transcription of stress-related genes [247]. Moreover, Yan et al. found that GhWRKY17 regulates plant sensitivity to drought by reducing ABA levels, and regulates the expression of ROS scavenging genes, such as *APX*, *CAT*, and *SOD*. In other words, GhWRKY17 responds to drought and salt stress by regulating the ABA signaling pathway, and ROS production in plant cells [248]. Similarly, GhWRKY68 responds to drought and salt stress by regulating ABA signaling and cellular ROS, too [249]. In addition, WRKY transcription factors also can participate in the process of stress resistance by regulating the expression of other transcription factors. Wei et al. proposed that two ERF family genes *NtERF5* and *NtEREBP-1* in transgenic plants overexpressing *TcWRKY53* were negatively induced, suggesting that *TcWRKY53* may regulate osmotic stress responses through interaction with ERF transcription factors rather than direct regulation of functional genes [250].

The NAC family of transcription factors is a class of plant-specific transcription factors with a variety of biological functions, which is characterized by highly conserved and specific NAC domains in the N-terminal of proteins. NAC plays an important role in plant resistance to drought stress by directly or by regulating the expression of genes involved in drought response. Fujita et al. indicated that RD26, as a dehydration-induced NAC protein, plays a transcriptional role in ABA-induced gene expression in plants under abiotic stress [251]. Moreover, Yong et al. showed that LlNAC2 was involved in DREB/CBF-COR and ABA signaling pathways to regulate stress tolerance in lily [252]. *Arabidopsis thaliana* with overexpression of *PwNAC2* exhibited greater drought tolerance by scavenging ROS, reducing membrane damage, slowing water loss, and increasing stomatal closure. In addition, the ABA or CBF pathway marker genes transgenic with the *PwNAC2* gene were significantly increased in Arabidopsis, suggesting that PwNAC2 enhanced plant tolerance to abiotic stress through multiple signaling pathways [253]. Jiang et al. proposed that RhNAC3, as a positive regulator, could improve the dehydration tolerance of rose petals mainly by regulating osmotic regulation-related genes [254]. In transgenic *Arabidopsis thaliana*, overexpression of *VvNAC17* enhanced drought resistance and upregulated expression of ABA and stress-related genes such as *ABI5*, *AREB1*, *COR15A*, *COR47*, *P5CS*, *RD22*, and *RD29A* [255]. However, excessive expression of stress-related genes may have negative effects on plant growth and development. Nakashima found that transgenic plants overexpressing *OsNAC6* improved drought, high salinity, and blast resistance, but resulted in dwarfing and low yield.

## 6. Conclusions

As mentioned above, in the past studies, the changes of plant external morphology and internal biochemical properties under drought stress have been described in detail. We have also gained a good understanding of signal transduction networks and molecular regulatory mechanisms in plants. Nevertheless, our current research is still incomplete and there are still many scientific problems to be solved. For example, ABA signaling networks are poorly described. In addition, although some famous abiotic stress-related gene families, such as AP2/ERF, MYB, NAC, etc., have been extensively studied by predecessors, there are still many unknown mechanisms in the large molecular regulation. Plants respond to water scarcity in different ways, and this is a complex process that we still need to work on unraveling. The research on the strategies of plants to cope with drought stress

can help us to better use scientific means to improve the adaptability of plants to water shortage environment and increase the yield of crops to play a more important role. Therefore, this review provides valuable background knowledge and theoretical basis for selective breeding, cross breeding, and molecular breeding of agricultural and forestry crops in the future by systematically analyzing and summarizing the mechanisms of plant response to drought.

**Author Contributions:** This review was mainly organized by X.Y., M.L., Y.W. (Yufei Wang), Y.W. (Yiran Wang), and Z.L., who were the contributors for collection of the materials and reversion of the manuscript. S.C. conceived the study and revised the manuscript. All authors have read and agreed to the published version of the manuscript.

**Funding:** This review was funded by the Fundamental Research Funds for the Central Universities (2572019CG08), the National Natural Science Foundation of China, grant number 31870659 and Heilongjiang Touyan Innovation Team Program (Tree Genetics and Breeding Innovation Team).

**Institutional Review Board Statement:** Not applicable.

**Informed Consent Statement:** Not applicable.

**Conflicts of Interest:** The authors declare no conflict of interest.

## References

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
