# Peer review of "Response Mechanism of Plants to Drought Stress"

_horticulturae, doi:10.3390/horticulturae7030050_

Round 1

Reviewer 1 Report

The reviewer had finished the review of the manuscript entitled “A Review on Response Mechanism of Plants to Drought Stress “. The manuscript is a review article covering drought stress-related genes/proteins and metabolites in plants. The authors provide an in-depth review of many important genes and proteins involved in drought stress response regulation. These include functional proteins (i.e. LEA, AQP) and regulatory proteins (i.e. CDPK, MAPK, TFs). There are comments to make the manuscript better:

  1. The title should be changed to “Response Mechanism of Plants to Drought Stress”.
  2. Apart from Figure 1, the reviewer encourage the authors provide one to two more figures to summarize all the related genes in different pathways in the manuscript.
  3. The authors should spell out full name of any abbreviation when first appear. For example, ABA, ROS in line 40 etc.
  4. In terms of proline metabolism and its relationship with osmotic stress, the authors may cite the following papers.
  5. Verslues PE1, Lasky JR1, Juenger TE, Liu T-W, Kumar MN (2014) Genome wide association mapping combined with reverse genetics identifies new effectors of low water potential-induced proline accumulation in Arabidopsis thaliana. Plant Physiology 164: 144-159      
  6. Verslues PE1, Lasky JR1, Juenger TE, Liu T-W, Kumar MN (2014) Genome wide association mapping combined with reverse genetics identifies new effectors of low water potential-induced proline accumulation in Arabidopsis thaliana. Plant Physiology 164: 144-159     
  7. Sharma S, Villamor JG, Verslues PE (2011) Essential role of tissue specific proline synthesis and catabolism in growth and redox balance at low water potential. Plant Physiology 157: 292-304
  8. Sharma S, Villamor JG, Verslues PE (2011) Essential role of tissue specific proline synthesis and catabolism in growth and redox balance at low water potential. Plant Physiology 157: 292-304
  9. In terms of AQP and its relationship with drought or salt stress, the authors may cite the following papers about AQP phosphorylation.
  10. Zhang S, Feng M, Chen W, Zhou X, Lu J, Wang Y, Li Y, Jiang CZ, Gan SS, Ma N, Gao J. In rose, transcription factor PTM balances growth and drought survival via PIP2;1 aquaporin. Nat Plants. 2019 Mar;5(3):290-299. doi: 10.1038/s41477-019-0376-1. Epub 2019 Mar 4
  11. Hsu JL, Wang LY, Wang SY, Lin CH, Ho KC, Shi FK, Chang IF. Functional phosphoproteomic profiling of phosphorylation sites in membrane fractions of salt-stressed Arabidopsis thaliana. Proteome Sci. 2009 Nov 10;7:42.
  12. Vialaret J, Di Pietro M, Hem S, Maurel C, Rossignol M, Santoni V. Phosphorylation dynamics of membrane proteins from Arabidopsis roots submitted to salt stress. Proteomics. 2014 May;14(9):1058-70.
  13. Prak S, Hem S, Boudet J, Viennois G, Sommerer N, Rossignol M, Maurel C, Santoni V. Share
  14. Multiple phosphorylations in the C-terminal tail of plant plasma membrane aquaporins: role in subcellular trafficking of AtPIP2;1 in response to salt stress. Mol Cell Proteomics. 2008 Jun;7(6):1019-30
  15. Chang IF, Hsu JL, Hsu PH, Sheng WA, Lai SJ, Lee C, Chen CW, Hsu JC, Wang SY, Wang LY, Chen CC. Comparative phosphoproteomic analysis of microsomal fractions of Arabidopsis thaliana and Oryza sativa subjected to high salinity. Plant Sci. 2012 Apr;185-186:131-42.
  16. In terms of drought-induced proteins, the authors may cite the following papers about Di19 proteins.
  17. Milla MA, Townsend J, Chang IF, Cushman JC. The Arabidopsis AtDi19 gene family encodes a novel type of Cys2/His2 zinc-finger protein implicated in ABA-independent dehydration, high-salinity stress and light signaling pathways. Plant Mol Biol. 2006 May;61(1-2):13-30.
  18. Li G, Tai FJ, Zheng Y, Luo J, Gong SY, Zhang ZT, Li XB. Two cotton Cys2/His2-type zinc-finger proteins, GhDi19-1 and GhDi19-2, are involved in plant response to salt/drought stress and abscisic acid signaling. Plant Mol Biol. 2010 Nov;74(4-5):437-52.
  19. Liu WX, Zhang FC, Zhang WZ, Song LF, Wu WH, Chen YF. Arabidopsis Di19 functions as a transcription factor and modulates PR1, PR2, and PR5 expression in response to drought stress. Mol Plant. 2013 Sep;6(5):1487-502
  20. Wu M, Liu H, Gao Y, Shi Y, Pan F, Xiang Y. The moso bamboo drought-induced 19 protein PheDi19-8 functions oppositely to its interacting partner, PheCDPK22, to modulate drought stress tolerance. Plant Sci. 2020 Oct;299:110605.
  21. Feng ZJ, Cui XY, Cui XY, Chen M, Yang GX, Ma YZ, He GY, Xu ZS. The soybean GmDi19-5 interacts with GmLEA3.1 and increases sensitivity of transgenic plants to abiotic stresses. Front Plant Sci. 2015 Mar 24;6:179.
  22. Qin LX, Li Y, Li DD, Xu WL, Zheng Y, Li XB. Arabidopsis drought-induced protein Di19-3 participates in plant response to drought and high salinity stresses. Plant Mol Biol. 2014 Dec;86(6):609-25.
  23. In terms of transcription factor in AP2/ERF family, the authors may cite the following papers.
  24. Cheng MC, Hsieh EJ, Chen JH, Chen HY, Lin TP. Arabidopsis RGLG2, functioning as a RING E3 ligase, interacts with AtERF53 and negatively regulates the plant drought stress response. Plant Physiol. 2012 Jan;158(1):363-75.
  25. Cheng MC, Liao PM, Kuo WW, Lin TP. The Arabidopsis ETHYLENE RESPONSE FACTOR1 regulates abiotic stress-responsive gene expression by binding to different cis-acting elements in response to different stress signals. Plant Physiol. 2013 Jul;162(3):1566-82.
  26. Hsieh EJ, Cheng MC, Lin TP. Functional characterization of an abiotic stress-inducible transcription factor AtERF53 in Arabidopsis thaliana. Plant Mol Biol. 2013 Jun;82(3):223-37.

Author Response

Point 1: The title should be changed to “Response Mechanism of Plants to Drought Stress”.

Response 1: Thanks a lot for the reviewer’s comments. We’ve changed the title to [Response Mechanism of Plants to Drought Stress].

Point 2: Apart from Figure 1, the reviewer encourage the authors provide one to two more figures to summarize all the related genes in different pathways in the manuscript.

Response 2: Thank you for your constructive suggestions. We agree with you very much. We reworked Figure 1 and added some content.

Point 3: The authors should spell out full name of any abbreviation when first appear. For example, ABA, ROS in line 40 etc.

Response 3: We are very sorry for our negligence of the explanation. We rechecked the full spelling of professional terms to make sure they had the full name when they first appeared. For example, in line 34 of page 1, we add and complete the full names of professional terms such as ABA and IP3.

Point 4: The reviewer suggested a number of constructive references.

Response 4: Thank you for the references, some of which are now included in the revised manuscript. Specific references are listed as follows:

Sharma, S., J. G. Villamor, and P. E. Verslues. "Essential Role of Tissue-Specific Proline Synthesis and Catabolism in Growth and Redox Balance at Low Water Potential." Plant Physiol 157, no. 1 (2011): 292-304

Zhang, S., M. Feng, W. Chen, X. Zhou, J. Lu, Y. Wang, Y. Li, and C. Z. Jiang. "In Rose, Transcription Factor Ptm Balances Growth and Drought Survival Via Pip2;1 Aquaporin." Nat Plants 5, no. 3 (2019): 290-99.

Cheng, M. C., P. M. Liao, W. W. Kuo, and T. P. Lin. "The Arabidopsis Ethylene Response Factor1 Regulates Abiotic Stress-Responsive Gene Expression by Binding to Different Cis-Acting Elements in Response to Different Stress Signals." Plant Physiol 162, no. 3 (2013): 1566-82.

Reviewer 2 Report

The review manuscript was focused on response mechanism of plants to drought stress. The paper has a significant contribution to the field. However, in order to increase its scientific soundness, I recommend several improvements:

- With such a long manuscript, it is necessary to add clear, colourful diagrams, charts, and figures. Graphic material will increase the readability of the article and it will be more interesting for readers,

- Figure 1 needs to be refined graphically, I suggest converting it to colour. Also it seems that the caption under the figure is too long,

- I recommend profound revision of English style and grammar throughout the manuscript,

- I suggest revision of the manuscript by increasing description of molecular aspects in the paper (e.g. gene expression studies in drought-exposed plants).

Author Response

Point 1: With such a long manuscript, it is necessary to add clear, colourful diagrams, charts, and figures. Graphic material will increase the readability of the article and it will be more interesting for readers.

Response 1: Thank you for your constructive suggestions. We agree with you very much. We reworked Figure 1 and added some content.

Point 2: Figure 1 needs to be refined graphically, I suggest converting it to colour. Also it seems that the caption under the figure is too long.

Response 2: Thank you very much for your constructive suggestions. We have remade Figure 1 according to your suggestions. The colors and forms of the picture are richer and we hope our modifications meet your expectations.

Point 3: I recommend profound revision of English style and grammar throughout the manuscript.

Response 3: Thanks for your constructive suggestion, which is highly appreciated. We have carefully scrutinized the manuscript, and made corresponding revisions including some typos, grammatical errors and incorrect use of articles, etc. we have uploaded a copy of the original manuscript with all the changes highlighted by using the track changes mode in MS Word. This manuscript has been revised extensively according to the reviewers' constructive suggestions. In addition, the expression of the manuscript has been improved with the help of a native English speaker.

Point 4: I suggest revision of the manuscript by increasing description of molecular aspects in the paper (e.g. gene expression studies in drought-exposed plants).

Response 4: Thank you for your comments. We have added a statement about molecules on page 24, line 1211.